# Disentangling Latent Risk Pathways via Bayesian Hypergraph Inference

**Shengxian Ding** [1]   **Haonan Gao** [1]   **Pangpang Liu** [1]   **Xinyuan Tian** [1]   **Yize Zhao** [1]

## Abstract

Electronic health records (EHR) pose large-scale multi-disease modeling problems in which many outcomes are rare and strongly influenced by shared risk factors. While modern approaches achieve strong predictive performance, they often treat diseases independently or rely on black-box architectures, offering limited insight into how risk factors organize disease risk and little principled uncertainty quantification. We introduce a Bayesian hypergraph inference framework that reframes multi-disease modeling around **latent, risk-factor-modulated disease pathways**. Risk factors act on hyperedges, latent disease subsets with shared risk patterns, allowing diseases to participate in multiple distinct pathways and enabling interpretable, higher-order structure beyond pairwise associations. A repulsion prior encourages parsimonious and identifiable structure, while posterior inference provides calibrated uncertainty over both disease groupings and risk-factor influence. To enable scalable inference on large EHR datasets, we develop a structured variational inference algorithm that preserves logical dependencies among hyperedge existence, disease membership, and pathway-level effects. Experiments on simulated data and the UK Biobank demonstrate stable and interpretable disease pathway structure, well-calibrated uncertainty, improved estimation for rare diseases, and competitive predictive performance.

## 1. Introduction

Electronic health records (EHR) enable large-scale modeling of disease risk across populations and many outcomes simultaneously. In this regime, individuals are often susceptible to multiple conditions, disease prevalence varies widely from common chronic disorders to rare diseases with few observed cases, and shared biological pathways, behavioral exposures, and social determinants induce complex dependencies across diseases (Goh et al., 2007; Skou et al., 2022). Understanding how disease risk is organized across populations is therefore central to epidemiology, precision medicine, and population health.

Crucially, these dependencies among diseases are **not uniform across risk factors**: different risk drivers organize diseases in different ways. For example, age may jointly increase risk for cardiovascular and metabolic diseases, while smoking predominantly affects respiratory and oncological conditions (Murray et al., 2020). These disease groupings are overlapping, inherently uncertain, and specific to the risk factor under consideration, reflecting distinct etiological patterns through which different exposures are associated with disease risk (Menche et al., 2015).

Modeling disease risk in this multi-disease setting poses two fundamental challenges. First, disease prevalence exhibits a pronounced long-tail structure: while some chronic conditions are common, most diseases are rare, providing insufficient data for stable independent estimation. Second, latent etiological structure is complex and higher-order: diseases can participate in multiple, risk-factor-specific pathways, requiring models to share information across outcomes without collapsing distinct latent mechanisms. A successful solution must therefore balance statistical efficiency with structured inductive bias, enabling interpretability through the attribution of risk while explicitly representing uncertainty arising from limited and heterogeneous data. The core challenge is not merely prediction, but **learning risk-factor–specific, overlapping latent structure over disease outcomes with uncertainty**.

Existing approaches address aspects of multi-disease modeling, but each falls short of this goal in critical ways. Independent disease-specific models, such as logistic regression, are transparent but treat diseases in isolation, ignoring shared structure and yielding poorly calibrated uncertainty for rare outcomes (King & Zeng, 2001; Heinze & Schemper, 2002). Conversely, multitask learning and joint modeling approaches improve robustness through information sharing (Caruana, 1997; Zhang & Yang, 2022), but typically rely on black-box architectures that entangle all risk fac-

[1]Department of Biostatistics, Yale University, New Haven, USA. Correspondence to: Yize Zhao <yize.zhao@yale.edu>.

*Proceedings of the 43$^{rd}$ International Conference on Machine Learning*, Seoul, South Korea. PMLR 306, 2026. Copyright 2026 by the author(s).

tors into a single latent space. This obscures the specific pathways through which different risk drivers exert their effects and provides limited insight into uncertainty over the learned structure. Finally, more structured probabilistic models capture outcome-level dependence (Blumm et al., 2009; Menche et al., 2015), but often focus on marginal correlations aggregated across all factors rather than disentangling risk-factor-specific disease organization, and can be difficult to scale to modern EHR datasets. As a result, existing methods cannot answer a central epidemiological question: **through which shared disease pathways does a given risk factor exert its effects, and how uncertain are we about that structure?**

To address this gap, we introduce a **Bayesian Hypergraph Pathway Inference (BHPI)** framework that reframes multi-disease modeling as the discovery of latent, risk-factor-modulated disease pathways (Figure 1). Our key insight is representational: while standard graphs and multitask models capture pairwise correlations or entangled shared effects, etiological pathways are inherently higher-order, involving groups of diseases rather than isolated pairs. We therefore model disease pathways as hyperedges in a latent hypergraph, where nodes represent diseases, and each hyperedge captures a shared pattern of risk-factor influence. Risk factors act directly on hyperedges to induce structured effects across groups of diseases. This decouples risk-factor influence from individual outcomes and naturally supports overlapping, factor-specific disease organization. Structurally, we introduce a repulsion prior that promotes parsimonious pathway discovery by discouraging redundant or highly overlapping hyperedges associated with the same risk factor. This prior mitigates latent structure collapse and plays a critical role in stabilizing inference in the long-tail regime, where weak signals from rare diseases would otherwise lead to degenerate solutions. Posterior inference in this model is challenging due to strong logical dependencies among hyperedge existence, disease membership, and hyperedge-level effects. Standard mean-field variational inference (VI) breaks these dependencies, leading to poorly calibrated uncertainty. We therefore develop a scalable structured VI algorithm that explicitly preserves these couplings, enabling uncertainty-aware learning of disease pathways and risk propagation in large EHR datasets.

We summarize our contributions as follows:

- We introduce a Bayesian hypergraph pathway inference framework with a repulsion prior, enabling parsimonious learning of risk-factor-specific, higher-order disease structure.

- We develop a scalable structured VI algorithm that preserves the logical dependencies (existence $\rightarrow$ membership $\rightarrow$ effect), overcoming known limitations of standard mean-field approximations.

- We validate the framework on synthetic data and the UK Biobank, demonstrating stable and interpretable pathway recovery, improved rare-disease risk estimation, and well-calibrated uncertainty.

## 2. Related Work

**Independent Multi-Disease Models**  A common baseline for multi-disease risk modeling is to train separate models for each outcome, typically using regularized logistic regression (Hoerl & Kennard, 1970; Tibshirani, 1996; Zou & Hastie, 2005). Specific penalized (Firth, 1993; King & Zeng, 2001; Heinze & Schemper, 2002) and Bayesian (Gelman et al., 2008) extensions further improve numerical stability in low-prevalence settings. But these approaches fundamentally treat diseases as independent prediction tasks, therefore failing to borrow statistical strength for rare outcomes.

**Multi-task Learning and Representation-based Models**  Multi-task learning (MTL) methods share information via common representations across related disease prediction tasks, using shared latent factors (Caruana, 1997; Evgeniou & Pontil, 2004; Argyriou et al., 2008; Standley et al., 2020), hierarchical priors (Bonilla et al., 2007), or neural network architectures (Miotto et al., 2016; Choi et al., 2016; Rajkomar et al., 2018; Harutyunyan et al., 2019; Rasmy et al., 2021) to jointly model multiple disease risks. While these approaches can implicitly capture commonalities across outcomes, the learned representations are typically implicit and black-box, obscuring the specific grouping of diseases and offering limited uncertainty quantification over the structure itself.

**Structured Outcome Learning**  A complementary line of work focuses on modeling statistical dependence among diseases through disease networks or comorbidity structures. Such models characterize patterns of disease co-occurrence in EHR data and have been used for descriptive analysis, risk stratification, and hypothesis generation (Blumm et al., 2009; Barabási et al., 2011; Jensen et al., 2014; Menche et al., 2015). However, these static networks aggregate all risk factors into a single correlation structure, failing to disentangle how different inputs drive different dependencies. Probabilistic approaches like multi-label learning (Tsoumakas & Vlahavas, 2007; Read et al., 2011) or correlated topic models (Lafferty & Blei, 2005; Li & McCallum, 2006) capture outcome dependence but often lack the feature-specific modulation required to understand etiology.

**Hypergraph Representation Learning**  Hypergraphs have emerged as a powerful tool for modeling higher-order correlations in data (Zhou et al., 2006; Feng et al., 2019; Yadati et al., 2019; Antelmi et al., 2023). Recent work in biomedical ML has applied hypergraphs to patient-concept

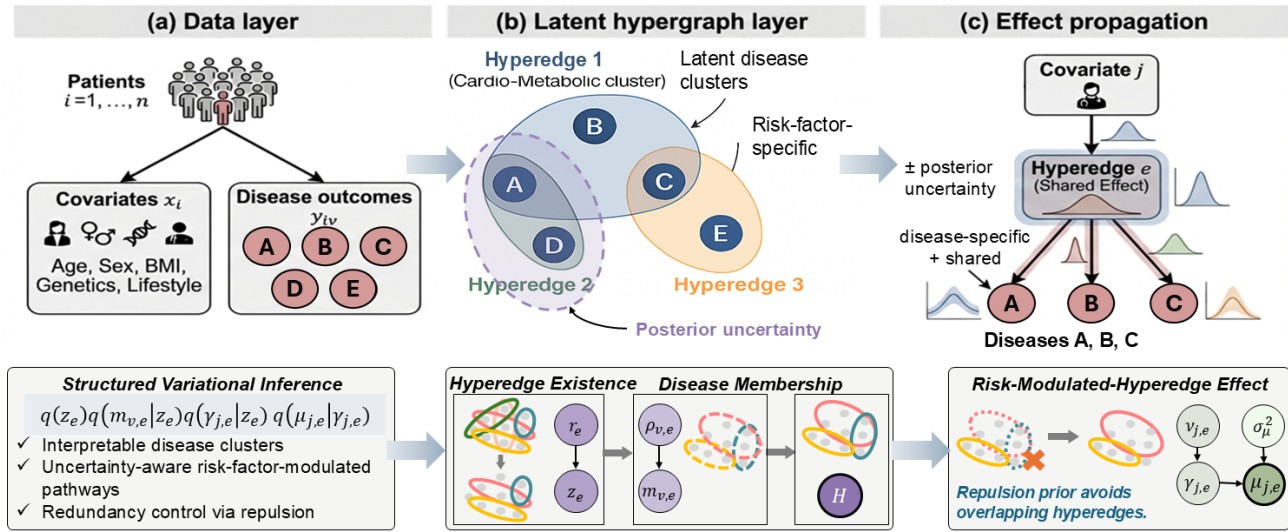

*Figure 1.* Bayesian Hypergraph Pathway Inference Workflow. (a) Input data consist of patient covariates and multiple disease outcomes. (b) A latent disease hypergraph models higher-order structure via hyperedges, allowing diseases to participate in multiple pathways with uncertain existence and membership. (c) Risk factors act on hyperedges to induce structured effects across diseases, yielding disentangled attribution. Shaded regions denote posterior uncertainty. Bottom panels: Structured Variational Inference (left) preserves logical dependencies to learn the latent topology including hyperedge existence and membership (middle), while a repulsion prior (right) discourages redundant pathways to ensure a parsimonious and interpretable latent structure.

interactions (Xu et al., 2023; Zhang et al., 2024). However, existing hypergraph neural networks generally assume the hypergraph structure is observed or inferred via heuristic similarity metrics. Our work differs by treating the hypergraph topology as a latent random variable to be inferred within a fully Bayesian generative framework, specifically designed to disentangle risk-factor modulation.

## 3. Bayesian Hypergraph Pathway Model

We propose a Bayesian hypergraph pathway model for learning **latent, higher-order output structure** in multi-disease prediction. The core object is a latent disease hypergraph whose hyperedges encode overlapping disease pathways, each selectively modulated by input features. Rather than modeling outcomes independently or through a single shared representation, our model learns which diseases are linked through which pathways, and which inputs activate each pathway, with calibrated uncertainty over both the latent structure and its effect propagation on outcomes.

### 3.1. Problem Setup

For each subject $i = 1, \ldots, N$, we observe a feature vector $\boldsymbol{x}_i \in \mathbb{R}^P$ and binary outcome $Y_{i,v} \in \{0, 1\}$ over a fixed set of $V$ diseases. Outcomes are defined only when the subject is at risk for disease $v$, indicated by $R_{i,v} = 1$, and all modeling and inference are restricted to valid subject-disease pairs. Our objectives are twofold: (i) to predict disease risk across many correlated outcomes, including low-

prevalence conditions; and (ii) to recover latent, overlapping disease pathways that explain how input features jointly influence multiple outcomes.

### 3.2. Observation Model

We link latent pathway structure to observed outcomes through a multivariate logistic observation model:

$$y_{i,v} \mid \tilde{\eta}_{i,v} \sim \text{Bernoulli}\left(\sigma\left(\tilde{\eta}_{i,v}\right)\right), \ \tilde{\eta}_{i,v} = \alpha_v + \boldsymbol{x}_i^\top \boldsymbol{\beta}_v, \ (1)$$

where $\alpha_v \sim \mathcal{N}(0, \varrho_v^2)$ is a disease-specific baseline risk and $\boldsymbol{\beta}_v \in \mathbb{R}^P$ captures risk-factor influences. Equation (1) serves as a measurement model linking latent structure to observed outcomes. Rather than estimating disease-specific effects $\boldsymbol{\beta}_v$ independently, we impose structured sharing of feature effects across diseases via a latent hypergraph, enabling information borrowing and interpretable higher-order representation over the output space.

### 3.3. Latent Hypergraph Representation

We represent latent disease pathways using a hypergraph $\mathcal{G} = (\mathcal{V}, \mathcal{E})$, where nodes $\mathcal{V} = \{1, \ldots, V\}$ correspond to diseases and each hyperedge $e \in \mathcal{E}$ represents shared response patterns to input features across subsets of outcomes. Hyperedges may overlap, allowing diseases to participate in multiple pathways. The hypergraph is encoded by an incidence matrix $H \in \{0, 1\}^{V \times E}$, where $H_{v,e} = \mathbb{I}\{\text{disease } v \text{ belongs to pathway } e\}$, and $E$ is a fixed upper bound on the number of hyperedges. This representation captures a higher-order output structure that

cannot be expressed using pairwise graphs or global shared representations.

### 3.4. Hypergraph-Induced Feature Effects

Disease-specific feature effects are induced by hyperedge-level effects:

$$\beta_{j,v} = d_v^{-1} \cdot \sum_{e=1}^{E} H_{v,e} \mu_{j,e}, \qquad (2)$$

where $\mu_{j,e}$ denotes the effect of risk factor $j$ on hyperedge $e$. The normalizing constant $d_v = E^{1/2}$ stabilizes the variance of induced effects and ensures risk magnitudes remain consistent as $E$ increases. This construction induces input-modulated, higher-order dependencies among outputs: a disease may be influenced by different features through distinct pathways, while a single feature may act on multiple disease subsets.

### 3.5. Hypergraph Structure Learning

Learning the hypergraph requires a combinatorial search over both pathway existence and disease membership. We adopt a hierarchical sparsity model in which each hyperedge $e$ has a binary existence indicator $z_e$, and conditional on existence, disease-level membership indicators $m_{v,e}$. The incidence matrix is defined as $H_{v,e} = z_e \cdot m_{v,e}$, with structured priors:

$$z_e \mid r_e \sim \text{Bernoulli}(r_e), \quad r_e \sim \text{Beta}(a_r, b_r),$$
$$m_{v,e} \mid z_e, \rho_{v,e} \sim (1 - z_e) \cdot \delta_0 + z_e \cdot \text{Bernoulli}(\rho_{v,e}),$$
$$\rho_{v,e} \sim \text{Beta}(a_\rho, b_\rho).$$

This hierarchy separates global hyperedge discovery from within-hyperedge disease composition, yielding parsimonious and interpretable latent structure.

### 3.6. Sparse Features-Modulated Hyperedge Effects

Not all features act on all hyperedges. For each feature $j$, we introduce a spike-and-slab prior over hyperedge-level effects:

$$\mu_{j,e} \mid \gamma_{j,e}, \sigma_\mu^2 \sim (1 - \gamma_{j,e})\delta_0 + \gamma_{j,e} \cdot \mathcal{N}(0, \sigma_\mu^2),$$
$$\sigma_\mu^2 \sim \text{Inv-Gamma}(a_\mu, b_\mu),$$

where $\gamma_{j,e}$ selects whether feature $j$ influences hyperedge $e$. To avoid degenerate solutions in which multiple hyperedges provide redundant explanations for the same feature, we introduce a repulsion prior that penalizes selecting highly overlapping hyperedges for a given feature $j$:

$$\mathcal{R}_{\text{rep}}(\gamma_j \mid H) \propto \exp\left\{-\lambda \sum_{e_1 < e_2} O(S_{e_1}, S_{e_2}) \cdot \gamma_{j,e_1} \gamma_{j,e_2}\right\},$$
$$O(S_{e_1}, S_{e_2}) = \frac{|S_{e_1} \bigcap S_{e_2}|}{\min\left(|S_{e_1}|, |S_{e_2}|\right)},$$

where $S_e = \{v : H_{v,e} = 1\}$, $\lambda$ controls repulsion strength, and $O(S_{e_1}, S_{e_2}) \in [0, 1]$ is the overlap coefficient between hyperedges $e_1$ and $e_2$, in which $O(S_{e_1}, S_{e_2}) = 1$ means complete overlaps. This encourages a disentangled and identifiable pathway structure within a single feature while allowing overlap hyperedges across different features. Moreover, we enforce the hierarchy $\gamma_{j,e} = 0$ whenever $z_e = 0$, ensuring that risk factors only select globally active hyperedges. Together, the joint prior on the risk-factor-modulated hyperedge effect selector is:

$$\gamma_j \mid \nu_j, H \sim \prod_{e=1}^{E} \left\{ z_e \cdot \text{Bernoulli}(\gamma_{j,e}; \nu_{j,e}) \cdot \mathcal{R}_{\text{rep}}(\gamma_j \mid H) \right.$$
$$\left. + (1 - z_e)\delta_0(\gamma_{j,e}) \right\},$$
$$\nu_{j,e} \sim \text{Beta}(a_\nu, b_\nu).$$

This hierarchy enforces the logical constraints: $z_e = 0 \Rightarrow m_{v,e} = \gamma_{j,e} = \mu_{j,e} = 0$, and $\gamma_{j,e} = 0 \Rightarrow \mu_{j,e} = 0$, ensuring coherent structure learning and improving identifiability without sacrificing predictive performance.

## 4. Scalable Structured Variational Inference

Posterior inference is challenging due to the combination of a non-conjugate likelihood and a combinatorial latent hypergraph with hard logical constraints linking discrete structure variables and continuous effects. To handle the non-conjugate logistic likelihood, we utilize Pólya–Gamma (PG) augmentation (Polson et al., 2013), introducing variables $\omega_{i,v}$ that render the likelihood conditionally Gaussian. Specifically, $\omega_{i,v} \sim \text{PG}(1, \tilde{\eta}_{i,v})$ yields a conditional log-likelihood quadratic in $\tilde{\eta}_{i,v}$, which enables closed-form CAVI updates (Appendix A). The hypergraph's logical constraints, however, induce strong posterior dependencies that violate the independence assumptions underlying standard mean-field factorization (e.g., $q(z_e) \prod_{v=1}^{V} q(m_{v,e})$), which assigns non-zero probability to configurations like $\{z_e = 0, m_{v,e} = 1\}$, a logical impossibility in our generative process. Preserving these dependencies is essential not only for uncertainty calibration but also for enforcing the repulsion-based identifiability constraints during inference.

### 4.1. Structured Variational Family

We propose a structured VI scheme that explicitly preserves the logical hierarchy while retaining computational tractability. We define the variational family as:

$$\mathcal{Q} = \prod_v q(\alpha_v) \prod_e q(z_e)q(r_e) \prod_{v,e} q(m_{v,e} \mid z_e)q(\rho_{v,e})$$
$$\prod_{j,e} q(\mu_{j,e} \mid \gamma_{j,e})q(\gamma_{j,e} \mid z_e)q(\nu_{j,e}) \prod_{i,v} q(\omega_{i,v})q(\sigma_\mu^2).$$

Key structural properties: (1) **Conditional Dependence:** the posteriors for disease membership $m_{v,e}$ and feature effects $\gamma_{j,e}, \mu_{j,e}$ are conditioned on the hyperedge existence $z_e$; (2) **Zero-Preservation:** if $q(z_e)$ concentrates on 0, the conditional factors $q(m_{v,e}|z_e = 0)$, $q(\gamma_{j,e} \mid z_e = 0)$ and $q(\mu_{j,e}|z_e = 0)$ collapse to Dirac deltas at 0, correctly removing inactive pathways consistently from the model. Overall, this structured variational formulation respects the hierarchical design of the generative model while remaining scalable to large EHR cohorts.

### 4.2. Optimization via Latent Variable Augmentation

We approximate the posterior by minimizing the Kullback-Leibler (KL) divergence between the variational family and the true posterior, equivalently maximizing the evidence lower bound (ELBO). Given the structural dependencies in BHPI, we employ Coordinate Ascent Variational Inference (CAVI, (Bishop & Nasrabadi, 2006)). While the variational family admits a product form, the optimal update for a factor $q_j(\boldsymbol{\theta}_j)$ must preserve conditional dependencies within the prior. The general update rule is:

$$q^*(\boldsymbol{\theta}_j) \propto \exp\left[\mathbb{E}_{q(\boldsymbol{\Theta}\setminus\boldsymbol{\theta}_j)}\left\{\log p(\boldsymbol{\Theta}, \mathcal{Y})\right\}\right].$$

For factors with internal logical dependencies, we use structured updates that account for the conditional entropy within the variational family; see Section A.3 for the full derivation. The variational posterior factors for BHPI take the following forms:

$$q(z_e) = \text{Bernoulli}(r_e^*),$$
$$q(m_{v,e} \mid z_e) = (1 - z_e)\delta_0(m_{v,e}) + z_e\text{Bernoulli}(\rho_{v,e}^*),$$
$$q(\gamma_{j,e} \mid z_e) = (1 - z_e)\delta_0(\gamma_{j,e}) + z_e\text{Bernoulli}\left(\nu_{j,e}^*\right),$$
$$q(\mu_{j,e} \mid \gamma_{j,e}) = (1 - \gamma_{j,e})\delta_0(\mu_{j,e}) + \gamma_{j,e}\mathcal{N}(\mu_{j,e}^*, \sigma_{j,e}^{2*}),$$
$$q(\alpha_v) = \mathcal{N}(\alpha_v^*, \varrho_v^{2*}), \quad q(\omega_{i,v}) = \text{PG}(1, \tilde{\eta}_{i,v}^*),$$
$$q(r_e) = \text{Beta}(a_e^{*(r)}, b_e^{*(r)}), \quad q(\rho_{v,e}) = \text{Beta}(a_{v,e}^{*(\rho)}, b_{v,e}^{*(\rho)}),$$
$$q(\nu_{j,e}) = \text{Beta}(a_{j,e}^{*(\nu)}, b_{j,e}^{*(\nu)}), \quad q(\sigma_\mu^2) = \text{Inv-Gamma}(a_\mu^*, b_\mu^*).$$

The quantities $r_e^*, \rho_{v,e}^*, \nu_{j,e}^*, \mu_{j,e}^*, \sigma_{j,e}^{2*}, \alpha_v^*, \varrho_v^{2*}, \tilde{\eta}_{i,v}^*, a_e^{*(r)}$, $b_e^{*(r)}, a_{v,e}^{*(\rho)}, b_{v,e}^{*(\rho)}, a_{j,e}^{*(\nu)}, b_{j,e}^{*(\nu)}, a_\mu^*, b_\mu^*$ are the variational parameters optimized to approximate the true posterior distribution. The optimized variational parameters aggregate evidence across the hierarchy. For instance, the update for $q(z_e)$ serves as a "global switch" that prunes redundant hyperedges by consolidating evidence from downstream disease-membership and feature-sparsity terms.

**Repulsion-Aware Variational Updates**  A unique challenge arises in updating the feature–hyperedge activation variables $\{\gamma_{j,e}\}$. Although $q(\gamma_{j,e} \mid z_e)$ is Bernoulli, its parameter $\nu_{j,e}^*$ is coupled across hyperedges by the repulsion

---

**Algorithm 1** Repulsion-Aware Coordinate Ascent (BHPI)

**Input:** Data $\mathcal{Y}$, max hyperedges $E$, repulsion strength $\lambda$.
**Initialize:** Variational parameters.
**while** ELBO not converged **do**
  // 1. Augmentation & Baseline Updates
  Update Pólya–Gamma variables $\tilde{\eta}_{i,v}^*$ and baseline intercepts $\alpha_v^*$, $\varrho_v^{2*}$ (Equations (9) and (10))
  // 2. Update Latent Risk-Hypergraph Structure
  **for** each hyperedge $e = 1, \ldots, E$ **do**
    Update risk-factor effects $\mu_{j,e}^*, \sigma_{j,e}^{2*}$ when $\gamma_{j,e} = 1$ (Equations (11) and (12))
    Compute expected overlap $\mathbb{E}_q\left[O(S_e, S_{e'}\right]$
    Update effect activation $\nu_{j,e}^*$ (Equation (13))
    Update disease memberships $\rho_{v,e}^*$ (Equation (14))
    Update hyperedge existence $r_e^*$ (Equation (15))
  **end for**
  // 3. Hyperparameter Update
  Update $a_e^{*(r)}, b_e^{*(r)}, a_{v,e}^{*(\rho)}, b_{v,e}^{*(\rho)}, a_{j,e}^{*(\nu)}, b_{j,e}^{*(\nu)}, a_\mu^*, b_\mu^*$
**end while**
**return** Optimized variational posterior distributions.

---

prior $\mathcal{R}_{\text{rep}}(\boldsymbol{\gamma}_j \mid H)$. This transforms the update into a competitive selection process where hyperedges associated with the same risk factor $j$ compete for activation. Under CAVI, the optimal log-odds for $\nu_{j,e}^*$ incorporates the expected log-prior contribution from all competing hyperedges $\{e' \neq e\}$. Concretely, we replace the discrete combinatorial overlap with its variational expectation $\mathbb{E}_q\left[O(S_e, S_{e'})\right]$. This term acts as a repulsive penalty, suppressing the simultaneous activation of multiple hyperedges that cover redundant disease pathways for the same feature, which ensures the discovery of a parsimonious and identifiable latent structure. The complete coordinate ascent procedure is summarized in Algorithm 1, with full mathematical derivations for each update rule provided in Section A.4. The per-iteration complexity of BHPI is approximately $\mathcal{O}(N \cdot E \cdot (P + V))$, ensuring that the computational cost scales linearly with the number of samples and the dimensions of the latent hypergraph.

## 5. Experiments

We evaluate the proposed BHPI framework to assess its ability to recover latent disease hypergraph structure modulated by shared risk factors, while maintaining accurate disease risk prediction. While predictive performance can be evaluated on held-out outcomes, assessment of latent hypergraph structure requires access to ground truth, which is unavailable in real-world EHR data. We therefore rely on simulations to evaluate both predictive accuracy and structural recovery under controlled conditions.[1]

---

[1] Our implementation is publicly available at https://github.com/Naomi-Ding/BHPI.

*Table 1.* Predictive performance on simulated data. Mean (standard deviation) AUC on held-out test sets ($\times$ 100). BHPI achieves competitive disease risk prediction while simultaneously inferring latent disease pathways.

| MODEL | $N = 2000$ | $N = 5000$ |
|---|---|---|
| BHPI | **75.00 (0.83)** | **74.63 (0.46)** |
| OPTIMAL LOGISTIC | 73.86 (0.83) | 74.15 (0.49) |
| LGBM | 68.50 (0.94) | 69.68 (0.56) |
| BINARY RELEVANCE | 71.50 (0.94) | 72.61 (0.64) |
| CLASSIFIER CHAIN | 71.25 (1.03) | 72.63 (0.63) |
| RAKELD | 70.99 (0.95) | 72.50 (0.60) |

*Table 2.* Recovery of latent risk structure on simulated data ($N = 2000$). Metrics include agreement between true and inferred disease-level risk contributions $\boldsymbol{\beta}$, AUC for recovering hypergraph structure $H$, and risk-factor-hyperedge associations $\boldsymbol{\gamma}$ ($\times$ 100).

| MODEL | COR$(\beta, \hat{\beta})$ | $H$-AUC | $\gamma$-AUC |
|---|---|---|---|
| BHPI | **97.85** | 96.96 | 97.85 |
| OPTIMAL LOGISTIC[2] | 95.65 | - | - |

## 5.1. Simulation Design and Baselines

We generate a latent disease hypergraph with $V = 30$ diseases and $E = 5$ hyperedges, where each hyperedge corresponds to a latent disease pathway driven by shared risk-factor influences. Diseases may participate in multiple hyperedges, reflecting overlapping risk mechanisms. Risk-factor influences on latent hypergraphs $\boldsymbol{\gamma} \in \mathbb{R}^{P \times E}$ are sparse: the first predictor affects multiple hyperedges, while each remaining predictor influences a single hyperedge. Nonzero influences are drawn from $\mathcal{N}(\mu, 0.5^2)$, with $\mu \in \{1, 1.5, 2\}$. Risk factors are independently sampled from a standard normal distribution. Disease outcomes are generated from the hypergraph-induced logistic model (Equations (1) and (2)). We evaluate sample sizes $N = 2000$ and $N = 5000$. For each setting, we perform 50 independent replicates using a 60/20/20 split for training, validation, and testing. Hyperparameters are selected on the validation set, and all reported predictive metrics are computed on held-out test data. We compare BHPI against (1) optimally tuned logistic regression (ridge, lasso, elastic-net), (2) LightGBM (LGBM), and (3) standard multi-label learning baselines: Binary Relevance, Classifier Chains, and RAkELd. These baselines represent independent, autoregressive, and higher-order label-interaction modeling strategies, respectively. While they are strong predictive competitors, they do not infer latent disease pathways or quantify uncertainty over shared mechanisms.

## 5.2. Predictive Performance and Structural Recovery

---

[2]$H$ and $\gamma$ recovery are not defined for logistic regression.

*Table 3.* Role of the repulsion prior in hypergraph identifiability ($N = 2000$). Predictive performance remains stable, while repulsion reduces redundancy and improves stability of inferred hyperedges, as measured by Jaccard similarity (Jacc.), overlap coefficients (Ov.) across replicates, effective hyperedge (Eff.#/RF) per risk factor (RF), and hyperedge overlap per RF (Ov./RF).

| $\lambda$ | AUC | JACC. | OV. | EFF.#/RF | OV./RF |
|---|---|---|---|---|---|
| $= 0$ | 74.69 | 0.542 | 0.800 | 1.992 | 0.097 |
| $> 0$ | **75.00** | **0.565** | **0.812** | **1.874** | **0.082** |

Table 1 shows that BHPI achieves predictive performance comparable to optimally tuned logistic regression and consistently outperforms tree-based and multi-label baselines. Despite its structured inductive bias, BHPI does not sacrifice predictive accuracy. The inferred disease-level risk contributions closely match the ground truth, with correlations exceeding 97%. Beyond prediction, BHPI accurately recovers latent hypergraph structure. The inferred disease–hyperedge incidence matrices closely match the ground truth, with AUCs exceeding 95% (Table 2). Risk-factor–hyperedge associations are also recovered with high fidelity, indicating that BHPI disentangles overlapping disease pathways rather than collapsing them into pairwise associations. In contrast, independent logistic regression cannot recover latent hypergraph structure by construction. A visualization of the true and inferred incidence matrices for a representative replicate is provided in Section B.1. This comparison demonstrates that the inferred hypergraph closely matches the true block structure with limited redundancy, identifying risk-factor-modulated disease pathways rather than collapsing them into pairwise associations.

## 5.3. Ablation of the Repulsion Prior

We examine the role of the repulsion prior by setting $\lambda = 0$, which allows a single risk factor to select highly overlapping hyperedges. Removing the repulsion prior has negligible impact on predictive accuracy, indicating that repulsion is not required for strong prediction. However, it substantially increases redundancy and instability in the inferred hypergraph, as reflected by reduced Jaccard similarity across replicates, greater hyperedge overlap, and a larger effective number of active hyperedges per risk factor (Table 3). These results demonstrate that the repulsion prior primarily improves identifiability and interpretability of latent disease pathways, rather than acting as a predictive regularizer.

## 5.4. Sensitivity Analysis

We evaluate the robustness of BHPI to the repulsion strength $\lambda$, and the assumed maximum number of latent hyperedges $E$. Across a wide range of $\lambda$ values, predictive performance remains stable, indicating that BHPI does not rely on fine-tuned regularization. Increasing $\lambda$ sharpens posterior inclu-

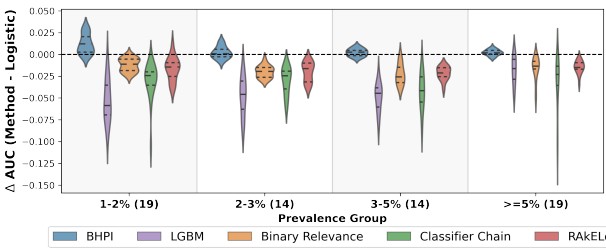

*Figure 2.* Predictive stability via structured information sharing. Distribution of per-disease $\Delta$AUC scores (relative to tuned Logistic Regression) stratified by disease prevalence. While baselines like LGBM and Classifier Chains exhibit high variance and instability for rare diseases, BHPI maintains robust performance by borrowing strength across shared latent pathways.

sion probabilities (PIPs) and reduces redundancy between inferred hyperedges, leading to more stable latent structures across random splits. We further examine robustness to over-specification of model capacity by varying $E$, which shows that even when $E$ is over-specified, the effective number of active hyperedges remains stable, demonstrating effective self-regularization. We view $E$ as a capacity upper bound. In practice, choose the smallest $E$ at which performance plateaus and the active edge count remains well below $E$. Detailed sensitivity curves and structural diagnostics are provided in Section B.3.

### 5.5. UK Biobank: Disentangling Real-World Disease Pathways

We evaluate BHPI on the UK Biobank (UKB), a large prospective cohort study linked to electronic health records (Bycroft et al., 2018). UKB serves as a challenging benchmark for multi-disease modeling due to its high-dimensional feature space, extreme class imbalance, and the complex dependency structure among chronic conditions. After quality control, the final analysis cohort comprises $N \approx 277,291$ participants. Additional dataset characteristics are summarized in Table 9. We consider $P = 71$ baseline risk factors spanning demographics, biomarkers, lifestyle exposures, and socioeconomic indicators. Disease outcomes comprise $V = 66$ chronic conditions derived from ICD-10 codes and mapped to Phecodes to improve phenotypic coherence and reduce coding noise (Wu et al., 2019; Wei et al., 2017). Disease prevalence ranges from common to extremely rare, placing the analysis firmly in the rare-disease regime. We set the maximum number of latent hyperedges to $E = 60$ to allow sufficient flexibility for capturing heterogeneous and overlapping disease pathways. Model performance is evaluated using stratified 60/20/20 train–validation–test splits, with results averaged over 100 independent random splits.

**Predictive Stability in the Rare-Disease Regime**  We first assess whether BHPI improves risk estimation without sacrificing predictive accuracy. Figure 2 reports per-disease

*Table 4.* Predictive calibration on UKB (66 diseases). BHPI ties for best Brier and achieves ECE $< 0.01$.

| MODEL | ECE | BRIER |
|---|---|---|
| BHPI | 0.005 | **0.041** |
| OPTIMAL LOGISTIC | **0.001** | **0.041** |
| LGBM | 0.021 | 0.043 |
| BINARY RELEVANCE | 0.031 | 0.045 |
| CLASSIFIER CHAIN | 0.044 | 0.052 |
| RAKELD | 0.006 | 0.042 |

$\Delta$AUC relative to optimally tuned logistic regression, stratified by disease prevalence. While performance on common diseases is largely saturated across all methods, a clear divergence emerges for low-prevalence conditions. Discriminative baselines, such as LightGBM and Classifier Chains, exhibit high-variance failure modes as evidenced by long lower tails in the $\Delta$AUC distribution. In contrast, BHPI consistently matches or exceeds the baseline, suggesting that BHPI stabilizes estimation for rare diseases by linking them to related conditions through shared latent structure.

Beyond discrimination, BHPI also achieves competitive predictive calibration across all baselines (Table 4). It ties for best Brier and achieves ECE $= 0.005$ ($< 0.01$); the small gap with Logistic is expected since Logistic optimizes per-disease calibration independently.

**Hypergraph-Specific Predictive Advantage**  To isolate the source of BHPI's advantage, we compare against two structured baselines (Table 5; mean $\Delta$AUC $\times 100$ vs per-disease logistic): Bayesian Factor Regression (BFR) with $K$ shared latent factors and per-disease loadings, which tests generic low-rank latent sharing; and a clique-expanded pairwise graph derived from the same discovered hyperedges, which tests whether pairwise reduction preserves the predictive signal. Only BHPI consistently outperforms optimally tuned logistic regression across prevalence groups, with the advantage strongest for rare diseases ($+1.2$, $p = 3 \times 10^{-5}$). BFR matches or underperforms logistic at every prevalence group, the best-performing clique-expanded variant achieves at most $+0.1$ in any group, and a learnable clique-GCN does not exceed chance (AUC $\approx 0.50$). The clique failure has a clear representational reading: the clique-expanded graph is $\sim 65\%$ dense even at PIP threshold 0.9, so clique expansion collapses higher-order group membership into near-uniform pairwise links. The primary advantage of the hypergraph representation is therefore *structural selectivity*: BHPI identifies which diseases co-participate in the same risk-modulated pathway, an assignment that clique expansion cannot recover by construction.

**Disentangling Risk-Modulated Pathways with Posterior Uncertainty**  To understand the structural mechanisms un-

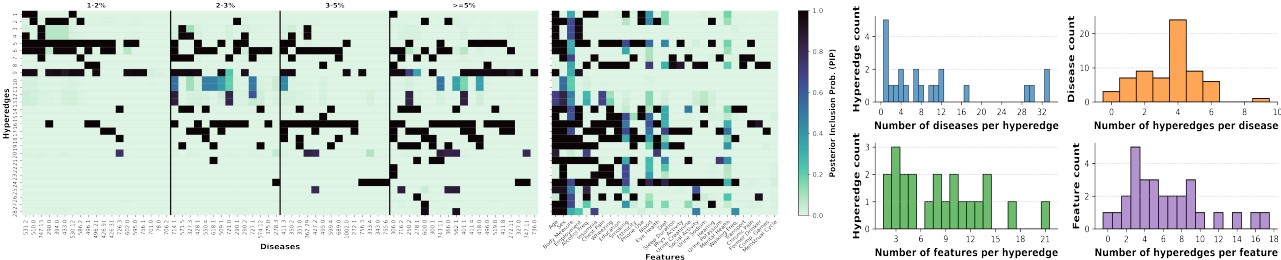

*Figure 3.* Disentanglement and parsimony of risk-modulated pathways. (Left) PIPs for disease–pathway membership and risk-factor activation, revealing sparse, overlapping disease organization and pathway-specific modulation with quantified uncertainty. (Right) Summary of the thresholded structure (PIP($z_e = 1$) > 0.5), showing compact pathway sizes, limited pathway participation per disease and risk factor, and overall structural parsimony.

*Table 5.* Mean $\Delta$AUC ($\times 100$) versus per-disease Logistic on UKB, with paired Wilcoxon $p$-values, stratified by disease prevalence. Only BHPI consistently outperforms Logistic. Bold indicates entries with $p < 0.05$.

| METHOD | $< 2\%$ | $2$–$5\%$ | $\geq 5\%$ |
|---|---|---|---|
| BHPI | **+1.2** (3E-5) | +0.2 (0.11) | **+0.2** (0.02) |
| BFR $K = 20$ | +0.1 (0.29) | **-0.3** (2E-6) | **-0.2** (4E-6) |
| BFR $K = 30$ | **-0.5** (1E-3) | **-0.8** (2E-8) | **-0.5** (4E-6) |
| CLIQUE-EXP. | **+0.1** (0.01) | **-0.2** (5E-4) | **-0.3** (4E-6) |

*Table 6.* PIP-stratified mean AUC ($\times 100$) on UKB (66 diseases) and simulation. High-PIP edges retain essentially all predictive signal; low-PIP edges are close to chance.

| CONFIGURATION | $< 2\%$ | $2$–$5\%$ | $\geq 5\%$ | SIMU |
|---|---|---|---|---|
| FULL MODEL | 68.9 | 69.7 | 71.6 | 75.7 |
| **HIGH-PIP ONLY** | **68.7** | **69.6** | **71.6** | **75.6** |
| RANDOM SUBSET | 60.2 | 61.3 | 62.7 | 65.7 |
| LOW-PIP ONLY | 58.6 | 55.0 | 57.4 | 51.2 |

derlying the observed predictive stability, we examine how BHPI disentangles latent disease pathways while enforcing parsimony. Figure 3(left) visualizes posterior uncertainty over the learned latent structure. The disease–hyperedge matrix exhibits a sparse, approximately block-structured pattern with overlapping memberships, indicating that diseases cluster into localized latent pathways while remaining allowed to participate in multiple contexts. In contrast, the risk-factor–hyperedge matrix reveals selective, pathway-specific activation patterns, highlighting that different predictors modulate distinct subsets of latent pathways rather than acting globally. Together, these patterns demonstrate that BHPI disentangles risk-factor–specific disease organization while explicitly quantifying uncertainty over both pathway composition and modulation.

As posterior mass concentrates, this uncertainty resolves into a compact global structure. Although initialized with $E = 60$ candidate hyperedges, the posterior concentrates on a parsimonious solution, with an expected 27.2 active hyperedges (28 with PIP > 0.5). As summarized in Figure 3(right), most hyperedges involve 2–12 diseases, while individual diseases participate in multiple pathways, reflecting heterogeneous risk contexts. On the predictor side, individual risk factors act on a limited number of pathways. This structured sparsity distinguishes BHPI from dense latent factor models that conflate unrelated mechanisms and hinder interpretability. At the global scale, Figure 4(left) provides a view of how risk factors propagate through latent pathways to affect multiple diseases. The resulting structure

is modular, overlapping and sparse, offering an interpretable summary of heterogeneous disease etiology learned from EHR data.

To verify that this discovered structure carries genuine predictive signal, we compare the full model against three masked variants: high-PIP edges only, low-PIP edges only, and a size-matched random subset (full per-prevalence breakdown in Table 6). The high-PIP edges retain essentially all predictive signal, while low-PIP edges are close to chance. This confirms that the posterior structure carries meaningful information rather than reflecting arbitrary uncertainty.

**Pathway-Level Illustration** To concretely demonstrate how our framework translates global posterior structure into interpretable mechanisms, we visualize representative latent pathways and their modulating risk factors in Figure 4 (right). This case study highlights three core properties of BHPI: *sparse disease membership*, *selective risk-factor activation*, and *graded posterior uncertainty* encoded directly through learned PIPs.

**(a) Divergent risk pathways (mechanism disentanglement):** A single risk factor may influence multiple, non-redundant disease pathways. For example, **Smoking** is inferred as a modulator for multiple non-redundant hyperedges. BHPI disentangles its impact into distinct biological contexts: a respiratory injury pathway grouping *Emphysema*, *Asthma*, *Chronic airway obstruction* and *Chronic bronchitis*, consistent with the well-documented smoking–

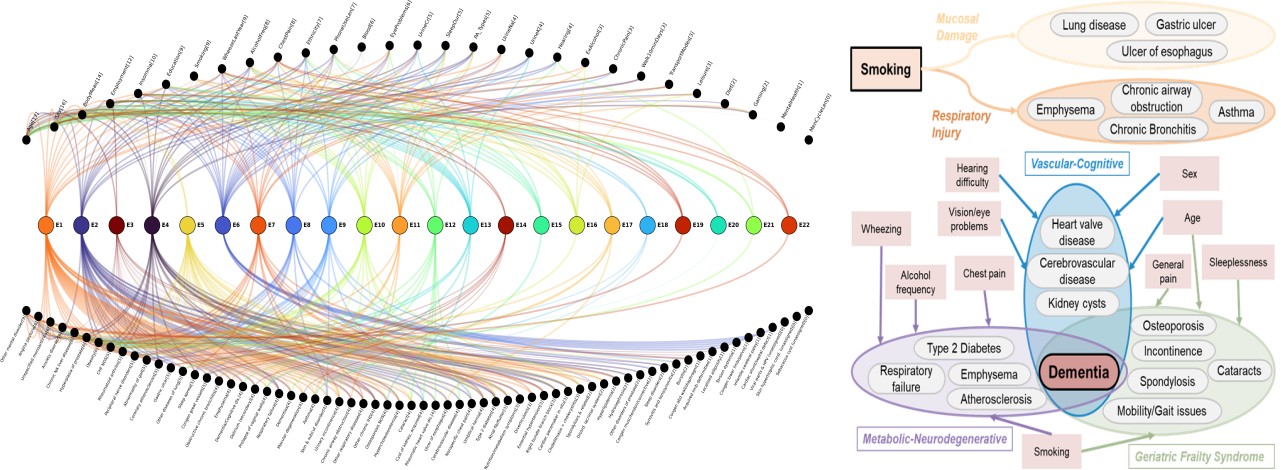

*Figure 4.* Global visualization of the learned risk-factor–to-disease pathway structure (hyperedges with PIP($z_e = 1$) > 0.5) (Left). Risk factors (top) connect to latent pathways (middle), which in turn connect to diseases (bottom). Edge widths correspond to PIPs, illustrating uncertainty, overlap, and modular risk propagation. Node annotations report the number of incident pathways. Pathway-level illustration (Right). Divergent risk pathways for smoking (top) and convergent disease pathways involving Dementia (bottom).

respiratory disease association (Forey et al., 2011), and a separate mucosal damage pathway linking *Ulcers*, which aligns with smoking as a peptic-ulcer risk factor (Kurata & Nogawa, 1997). The repulsion prior prevents these pathways from collapsing into a single dense factor, avoiding the entanglement typical of standard multitask models.

**(b) Convergent disease contexts (heterogeneity discovery):** Conversely, BHPI captures disease heterogeneity by allowing a single disease to participate in multiple pathways. For instance, **Dementia** belongs simultaneously to: (i) Vascular-cognitive pathways: shared with *Cerebrovascular disease* and *Heart valve disease*, modulated by *Age*, *Sex*, and sensory impairment, indicating vascular etiology and supported by the vascular cognitive impairment literature (Van Der Flier et al., 2018); (ii) Geriatric frailty pathway: grouped with *Osteoporosis*, *Incontinence*, and *Mobility issues*, driven by *Age*, *General pain*, reflecting systemic decline and frailty–dementia clustering (Kojima et al., 2016); (iii) Metabolic-neurodegenerative pathway: linked with *Type 2 Diabetes* and *Atherosclerosis*, modulated by *Alcohol frequency* and *Smoking*, consistent with metabolic exhaustion mechanisms and the diabetes–dementia link (Xue et al., 2019). Independent multimorbidity clustering studies also report closely related respiratory, cardiometabolic, neuropsychiatric, and dementia-linked patterns (Grande et al., 2021; Simões et al., 2017). These overlapping memberships demonstrate how BHPI represents heterogeneous disease etiology without enforcing mutually exclusive clustering.

## 6. Conclusion and Discussion

We introduced **Bayesian Hypergraph Pathway Inference (BHPI)**, a structured framework for multi-disease modeling

that discovers latent, risk-factor–modulated disease pathways. By representing pathways as hyperedges and allowing risk factors to act directly on this structure, BHPI captures higher-order and overlapping disease organization that is inaccessible to pairwise or independent outcome models. Across simulations and the UK Biobank, BHPI demonstrates stable prediction in the rare-disease regime, identifies parsimonious and non-redundant pathways via a repulsion prior, and provides calibrated posterior uncertainty over both pathway composition and risk-factor modulation through structured variational inference, offering interpretability beyond black-box architectures.

Several extensions warrant future investigation. Incorporating longitudinal EHR data would allow BHPI to model disease trajectories and time-varying pathway activation. Scaling inference to high-dimensional genetic features may require stochastic or amortized variants of the current framework. Although BHPI focuses on associative structure learning, integrating causal constraints or Mendelian randomization could further strengthen causal interpretability. Finally, while the linear observation model preserves interpretability and closed-form CAVI updates, coupling the hypergraph structure with nonlinear predictors is a promising extension.

## Acknowledgements

This research has been conducted using the UK Biobank Resource under application number 176775. This work was supported by National Institutes of Health (NIH) under awards R01AG081413, R01EB034720, R01AG068191. We thank the anonymous reviewers for their constructive feedback, which substantially improved the paper.

## Impact Statement

This paper presents work whose goal is to advance the field of Machine Learning. There are many potential societal consequences of our work, none which we feel must be specifically highlighted here.

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

# A. Variational Inference Details

In this section, we provide the full derivation of the Coordinate Ascent Variational Inference (CAVI) updates summarized in the main text. We begin by defining the complete log-joint probability of the generative model and the variational family.

## A.1. Pólya–Gamma Augmentation

To handle the non-conjugate logistic likelihood, we introduce auxiliary variables $\omega_{i,v} \mid \tilde{\eta}_{i,v} \sim$ Pólya–Gamma$(1, \tilde{\eta}_{i,v})$. This transforms the conditional posterior of the log-odds into a Gaussian form.

## A.2. Full Joint Posterior

Let $\boldsymbol{\Theta} = \left\{ \boldsymbol{z}(\mathbb{R}^E), \boldsymbol{r}(\mathbb{R}^E), \boldsymbol{m}(\mathbb{R}^{V \times E}), \boldsymbol{\rho}(\mathbb{R}^{V \times E}), \boldsymbol{\mu}(\mathbb{R}^{p \times E}), \boldsymbol{\gamma}(\mathbb{R}^{p \times E}), \boldsymbol{\nu}(\mathbb{R}^{p \times E}), \sigma_{\mu}^2, \boldsymbol{\alpha} \right\}$ denote the full set of model parameters, and the observed data is $\mathcal{Y}$. For each subject $i$, the diagnosis status for disease $v$ is $y_{i,v} \in \{0, 1\}$ if the disease is observed after the baseline, otherwise $y_{i,v} = $ NA. To eliminate the effect of prior-baseline diagnosis, we define a risk indicator $R_{i,v} = \mathbb{I}(\text{no diagnosis of } v \text{ before baseline})$. Then the disease outcome is only defined when $R_{i,v} = 1$. Conditional on $\boldsymbol{\omega}$ where $\omega_{i,v} \sim$ Pólya–Gamma$(1, 0)$, the log-likelihood becomes

$$\log p(\mathcal{Y} \mid \boldsymbol{\Theta}, \boldsymbol{\omega}) \propto \sum_{i,v} R_{i,v} \left[ \kappa_{i,v} \tilde{\eta}_{i,v} - \frac{1}{2} \omega_{i,v} \tilde{\eta}_{i,v}^2 \right], \quad \text{where} \quad \kappa_{i,v} = y_{i,v} - 1/2. \tag{3}$$

The posterior distribution of the model parameters is expressed as

$$p(\boldsymbol{\Theta} \mid \mathcal{Y}, \boldsymbol{\omega}) \propto p(\mathcal{Y} \mid \boldsymbol{\Theta}, \boldsymbol{\omega}) p(\boldsymbol{\Theta}),$$

where $p(\boldsymbol{\Theta})$ is the prior distribution over all parameters. Figure 5 gives the full probabilistic graphical model corresponding to the generative process introduced in Section 3.3 of the main text. Thus, the posterior kernel is given by

$$p(\boldsymbol{\Theta} \mid \mathcal{Y}, \boldsymbol{\omega}) \propto p(\mathcal{Y} \mid \boldsymbol{\Theta}, \boldsymbol{\omega}) \cdot \prod_{e=1}^{E} \prod_{v=1}^{V} p(m_{v,e} \mid \rho_{v,e}, z_e) p(\rho_{v,e}) \cdot \prod_{e=1}^{E} p(z_e \mid r_e) p(r_e)$$
$$\cdot \prod_{j=1}^{p} \prod_{e=1}^{E} p(\mu_{j,e} \mid \gamma_{j,e}, \sigma_{\mu}^2) \cdot \prod_{j=1}^{p} p(\boldsymbol{\gamma}_j \mid \boldsymbol{z}, \boldsymbol{m}, \boldsymbol{\nu}_j) \cdot \prod_{j=1}^{p} \prod_{e=1}^{E} p(\nu_{j,e}) \cdot p(\sigma_{\mu}^2). \tag{4}$$

The log posterior is

$$\log p(\boldsymbol{\Theta} \mid \mathcal{Y}, \boldsymbol{\omega}) \propto \log p(\mathcal{Y} \mid \boldsymbol{\Theta}, \boldsymbol{\omega}) + \sum_{e=1}^{E} \sum_{v=1}^{V} \log p(m_{v,e} \mid \rho_{v,e}, z_e)$$
$$+ \sum_{e=1}^{E} \log p(z_e \mid r_e) + \sum_{j=1}^{p} \sum_{e=1}^{E} \log p(\mu_{j,e} \mid \gamma_{j,e}, \sigma_{\mu}^2) + \sum_{j=1}^{p} \log p(\boldsymbol{\gamma}_j \mid \boldsymbol{z}, \boldsymbol{m}, \boldsymbol{\nu}_j)$$
$$+ \sum_{e=1}^{E} \sum_{v=1}^{V} \log p(\rho_{v,e}) + \sum_{e=1}^{E} \log p(r_e) + \sum_{j=1}^{p} \sum_{e=1}^{E} \log p(\nu_{j,e}) + \log p(\sigma_{\mu}^2), \tag{5}$$

$$\text{with} \quad \log p(\boldsymbol{\gamma}_j \mid \boldsymbol{z}, \boldsymbol{m}, \boldsymbol{\nu}_j) = \sum_{e=1}^{E} z_e \left[ \gamma_{j,e} \log \nu_{j,e} + (1 - \gamma_{j,e}) \log(1 - \nu_{j,e}) \right]$$
$$- \omega_\gamma \sum_{e_1 < e_2} O(H_{e_1}, H_{e_2}) \cdot z_{e_1} \gamma_{j,e_1} \cdot z_{e_2} \gamma_{j,e_2}.$$

## A.3. Optimal Updating Rule under Coordinate Ascent

Since the posterior in (4) is computationally intractable, we can use the variational inference (VI) framework (Blei et al., 2017), which approximates the posterior distribution of the model parameters by a simpler distribution. To develop an

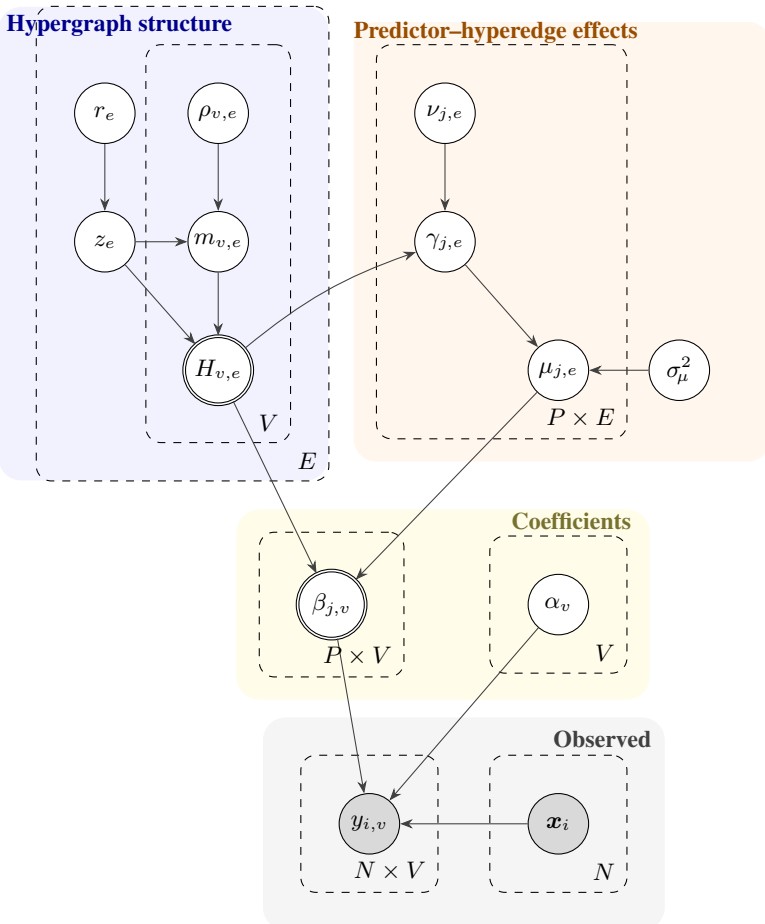

*Figure 5.* **Plate diagram of the BHPI generative model.** The generative DAG is: $z_e \to m_{v,e}$; $H_{v,e} = z_e m_{v,e}$; $\{z_e, H\} \to \gamma_{j,e} \to \mu_{j,e}$; $\beta_{j,v} = d_v^{-1} \sum_e H_{v,e} \mu_{j,e}$; $y_{i,v} \sim \text{Bernoulli}(\sigma(\alpha_v + x_i^\top \beta_v))$.

efficient VI algorithm, we first approximate the posterior by minimizing the Kullback-Leibler (KL) divergence between a variational family $q(\mathbf{\Theta})$ and the true posterior, i.e.,

$$q^*(\mathbf{\Theta}) = \arg \min_{q(\mathbf{\Theta}) \in \mathcal{Q}} \text{KL}\left(q(\mathbf{\Theta}) || p(\mathbf{\Theta} \mid \mathcal{Y})\right),$$

$$\text{where KL}\left(q(\mathbf{\Theta}) || p(\mathbf{\Theta} \mid \mathcal{Y})\right) = \mathbb{E}_{q(\mathbf{\Theta})}\left[\log q(\mathbf{\Theta})\right] - \mathbb{E}_{q(\mathbf{\Theta})}\left[\log p(\mathbf{\Theta} \mid \mathcal{Y})\right]$$
$$= \mathbb{E}_{q(\mathbf{\Theta})}\left[\log q(\mathbf{\Theta})\right] - \mathbb{E}_{q(\mathbf{\Theta})}\left[\log p(\mathbf{\Theta}, \mathcal{Y})\right] + \log p(\mathcal{Y}).$$

Recall that the evidence lower bound (ELBO) is defined as

$$\mathcal{L}(q) = \mathbb{E}_{q(\mathbf{\Theta})}\left[\log p(\mathbf{\Theta}, \mathcal{Y})\right] - \mathbb{E}_{q(\mathbf{\Theta})}\left[\log q(\mathbf{\Theta})\right] = \log p(\mathcal{Y}) - \text{KL}\left(q(\mathbf{\Theta}) || p(\mathbf{\Theta} \mid \mathcal{Y})\right), \quad (6)$$

so that minimizing this KL divergence is equivalent to maximizing the ELBO.

To present the derivation of the update rule of our proposed structured VI scheme, which preserves the logical dependencies emphasized in Section 4.2, we start with the generic mean-field variational inference (MFVI) (Blei et al., 2017), which assumes mutual independence across each parameter, to express the variational distribution $q(\mathbf{\Theta})$ in a factorized form, i.e., $q(\mathbf{\Theta}) = \prod_j q_j(\theta_j)$. Therefore, we can find $q^*(\mathbf{\Theta})$ using a coordinate ascent variational inference (CAVI) (Bishop & Nasrabadi, 2006) to iteratively optimize each factor of the mean-field variational distribution while holding the others fixed, until a convergence criterion is met. That is, the optimal $q_j(\theta_j)$ under MFVI is given by maximizing the ELBO in Equation

(6) in terms of the factor $\theta_j$, i.e., $q_j^*(\theta_j) = \arg\max_{q_j(\theta_j)} \mathcal{L}(q_j)$, where

$$
\begin{aligned}
\mathcal{L}(q_j) &= \mathbb{E}_{q_j(\theta_j)} \left[ \mathbb{E}_{q_{-j}(\Theta_{-j})} \{ \log p(\theta_j, \Theta_{-j}, \mathcal{Y}) \} \right] - \mathbb{E}_{q_j(\theta_j)} \left[ \log q_j(\theta_j) \right] \\
&= \mathbb{E}_{q_j(\theta_j)} \left[ \log \left( \exp \left[ \mathbb{E}_{q_{-j}(\Theta_{-j})} \{ \log p(\theta_j, \Theta_{-j}, \mathcal{Y}) \} \right] \right) \right] - \mathbb{E}_{q_j(\theta_j)} \left[ \log q_j(\theta_j) \right] \\
&= \mathbb{E}_{q_j(\theta_j)} \left[ \log \tilde{p} \left( \theta_j \mid \Theta_{-j}, \mathcal{Y} \right) \right] - \mathbb{E}_{q_j(\theta_j)} \left[ \log q_j(\theta_j) \right] + \log(\text{const}) \\
&= -\mathrm{KL} \left( q_j(\theta_j) \| \tilde{p} \left( \theta_j \mid \Theta_{-j}, \mathcal{Y} \right) \right) + \log(\text{const}),
\end{aligned}
$$

$$
\tilde{p}(\theta_j \mid \Theta_{-j}, \mathcal{Y}) = \frac{\exp \left[ \mathbb{E}_{q_{-j}(\Theta_{-j})} \{ \log p(\theta_j, \Theta_{-j}, \mathcal{Y}) \} \right]}{\text{const}},
$$

$\Theta_{-j}$ denotes all variables in $\Theta$ except $\theta_j$, $q_{-j}(\Theta_{-j}) = \prod_{k \neq j} q_k(\theta_k)$, and const is a normalization term irrelevant to $q_j(\theta_j)$. So the optimal update for $q_j(\theta_j)$ is obtained by minimizing the KL divergence between $q_j(\theta_j)$ and $\tilde{p}(\theta_j \mid \Theta_{-j}, \mathcal{Y})$, that is,

$$
q_j^*(\theta_j) = \arg\min_{q_j(\theta_j)} \mathrm{KL} \left( q_j(\theta_j) \| \tilde{p} \left( \theta_j \mid \Theta_{-j}, \mathcal{Y} \right) \right) = \tilde{p} \left( \theta_j \mid \Theta_{-j}, \mathcal{Y} \right) \propto \exp \left[ \mathbb{E}_{q_{-j}(\Theta_{-j})} \{ \log p(\Theta, \mathcal{Y}) \} \right]. \tag{7}
$$

By matching $q_j(\theta_j)$ to the conditional distribution of $\theta_j$ given the observed data $\mathcal{Y}$ and the other factors, this update rule (7) makes $q_j^*(\theta_j)$ as close to the true posterior as possible within the distribution family.

In our structured variational family, certain variables are modeled via conditional distributions (e.g., $q(m_{v,e}, z_e) = q(m_{v,e} \mid z_e) q(z_e)$). To derive the optimal update for the marginal factor $q(\theta_j)$, we isolate the terms in the ELBO that depend on $\theta_j$. Unlike the standard mean-field case, the entropy term $\mathbb{E}_q[\log q]$ now contributes a conditional entropy component, leading to the update rule:

$$
q^*(\boldsymbol{\theta}_j) \propto \exp \left[ \mathbb{E}_{q(\Theta \setminus \boldsymbol{\theta}_j)} \{ \log p(\Theta, \mathcal{Y}) - \log q(\Theta_{-j} \mid \boldsymbol{\theta}_j) \} \right], \tag{8}
$$

## A.4. Coordinate Ascent Updates for BHPI

### A.4.1. FROM OPTIMAL FACTORS TO ITERATIVE UPDATES.

Equations (9)–(15) give the optimal coordinate-wise variational factors under CAVI. In implementation, the corresponding variational parameters are updated by substituting the current expectations of the other factors into these expressions at each iteration, as summarized in Algorithm 1. For example, the update for $q(z_e)$ takes current estimates of $q(m_{v,e})$ and $q(\gamma_{j,e})$ as inputs and outputs a closed-form Bernoulli parameter, exactly the substitution that Algorithm 1 executes at each coordinate step.

**Pólya–Gamma augmented variables**

$$
q(\omega_{i,v}) = \text{Pólya–Gamma}(1, \tilde{\eta}_{i,v}^*), \quad \text{where } \tilde{\eta}_{i,v}^* = \sqrt{\mathbb{E}_q \left[ \tilde{\eta}_{i,v}^2 \right]}. \tag{9}
$$

**Disease-specific intercepts** $\quad q^*(\alpha_v) = \mathcal{N}(\alpha_v^*, \varrho_v^{2*})$, where

$$
\varrho_v^{2*} = \left[ \sum_{i=1}^N R_{i,v} \mathbb{E}_q[\omega_{i,v}] + \frac{1}{\sigma_\alpha^2} \right]^{-1}, \quad \alpha_v^* = \varrho_v^{2*} \left\{ \sum_{i=1}^N R_{i,v} \left( \kappa_{i,v} - \mathbb{E}_q[\omega_{i,v}] \mathbb{E}_q[\eta_{i,v}] \right) \right\}. \tag{10}
$$

**Risk-factor-hyperedge effects** $\quad q^*(\mu_{j,e} \mid \gamma_{j,e}) = (1 - \gamma_{j,e}) \cdot \delta_0 + \gamma_{j,e} \cdot \mathcal{N}(\mu_{j,e}^*, \sigma_{j,e}^{2*})$, where

$$
\sigma_{j,e}^{2*} = \left( \mathbb{E}_q \left[ \sigma_\mu^{-2} \right] + \sum_{i=1}^N \sum_{v=1}^V R_{i,v} x_{i,j}^2 \mathbb{E}_q[\omega_{i,v}] \mathbb{E}_q \left[ s_{v,e}^2 \mid z_e = 1 \right] \right)^{-1}, \quad s_{v,e} = d_v^{-1} H_{v,e}, \quad a_{i,v}^{\setminus(j,e)} = \eta_{i,v} - x_{i,j} s_{v,e} \mu_{j,e}, \tag{11}
$$

$$
\mu_{j,e}^* = \sigma_{j,e}^{2*} \left\{ \sum_{i=1}^N \sum_{v=1}^V R_{i,v} x_{i,j} \left( \kappa_{i,v} - \mathbb{E}_q[\omega_{i,v}] \left\{ \mathbb{E}_q \left[ a_{i,v}^{\setminus(j,e)} \mid z_e = 1 \right] + \alpha_v^* \right\} \right) \mathbb{E}_q[s_{v,e} \mid z_e = 1] \right\}. \tag{12}
$$

**Risk-factor-hyperedge effect selector** $\quad q(\gamma_{j,e} \mid z_e) = (1 - z_e) \cdot \delta_0 + z_e \cdot \text{Bernoulli}(\nu_{j,e}^*)$, where

$$
\text{logit}(\nu_{j,e}^*) = \frac{1}{2} \left\{ \frac{\left( \mu_{j,e}^* \right)^2}{\sigma_{j,e}^{2*}} + \log(\sigma_{j,e}^{2*}) - \mathbb{E}_q \left[ \log \sigma_\mu^2 \right] \right\} + \mathbb{E}_q \left[ \log \frac{\nu_{j,e}}{1 - \nu_{j,e}} \right] - \lambda \sum_{e' \neq e} \mathbb{E}_q \left[ O(S_e, S_{e'}) \right] r_{e'} \nu_{j,e'}. \tag{13}
$$

**Node-hyperedge incidence** $q(m_{v,e} \mid z_e) = (1 - z_e) \cdot \delta_0 + z_e \cdot \text{Bern}(\rho_{v,e}^*)$, where

$$\text{logit}(\rho_{v,e}^*) = \mathbb{E}_q \left[ \log \frac{\rho_{v,e}}{1 - \rho_{v,e}} \right] + d_v^{-1} \mathbb{E}_q \left[ \zeta_{v,e} \mid z_e = 1 \right] - \frac{1}{2} d_v^{-2} \mathbb{E}_q \left[ \xi_{v,e} \mid z_e = 1 \right]$$

$$- \lambda \sum_{j=1}^{P} \nu_{j,e} \sum_{e' \neq e} \mathbb{E}_q \left[ O \left( S_e, S_{e'} \mid m_{v,e} = 1 \right) - O \left( S_e, S_{e'} \mid m_{v,e} = 0 \right) \right] r_{e'} \nu_{j,e'}, \tag{14}$$

$$\zeta_{v,e} = \sum_{i=1}^{N} R_{i,v} \left( \kappa_{i,v} - \omega_{i,v} \left[ a_{i,v}^{(-e)} + \alpha_v \right] \right) b_{i,e}, \;\; \xi_{v,e} = \sum_{i=1}^{N} R_{i,v} \omega_{i,v} b_{i,e}^2, \;\; b_{i,e} = \boldsymbol{x}_i^\top \boldsymbol{\mu}_e, \;\; a_{i,v}^{(-e)} = \eta_{i,v} - s_{v,e} b_{i,e}.$$

**Hyperedge existence** $q(z_e) = \text{Bernoulli}(r_e^*)$, where

$$\text{logit}(r_e^*) = \sum_{v=1}^{V} \rho_{v,e}^* \left( d_v^{-1} \mathbb{E}_q \left[ \zeta_{v,e} \mid z_e = 1 \right] - \frac{1}{2} d_v^{-2} \mathbb{E}_q \left[ \xi_{v,e} \mid z_e = 1 \right] \right) + \mathbb{E}_q \left[ \log \frac{r_e}{1 - r_e} \right]$$

$$+ \sum_{v=1}^{V} \left\{ \rho_{v,e}^* \left[ \mathbb{E}_q \left[ \log \rho_{v,e} \right] - \log \rho_{v,e}^* \right] + (1 - \rho_{v,e}^*) \left[ \mathbb{E}_q \left[ \log(1 - \rho_{v,e}) \right] - \log(1 - \rho_{v,e}^*) \right] \right\}$$

$$+ \sum_{j=1}^{P} \left\{ \nu_{j,e}^* \left[ \mathbb{E}_q \left[ \log \nu_{j,e} \right] - \log \nu_{j,e}^* \right] + (1 - \nu_{j,e}^*) \left[ \mathbb{E}_q \left[ \log(1 - \nu_{j,e}) \right] - \log(1 - \nu_{j,e}^*) \right] \right\}$$

$$- \lambda \sum_{j=1}^{P} \nu_{j,e} \sum_{e' \neq e} \mathbb{E}_q \left[ O(S_e, S_{e'}) \right] r_{e'} \nu_{j,e'}. \tag{15}$$

**Slab variance** $q(\sigma_\mu^2) = \text{Inverse-Gamma} \left( a_\mu^*, \; b_\mu^* \right)$, where $a_\mu^* = a_\mu + \frac{1}{2} \sum_{j=1}^{P} \sum_{e=1}^{E} r_e^* \nu_{j,e}^*$, and $b_\mu^* = b_\mu + \frac{1}{2} \sum_{j=1}^{P} \sum_{e=1}^{E} r_e^* \nu_{j,e}^* \left[ \left( \mu_{j,e}^* \right)^2 + \sigma_{j,e}^{2*} \right]$.

**Prior inclusion probabilities**

$$q(\nu_{j,e}) = \text{Beta}(a_{j,e}^{*(\nu)}, b_{j,e}^{*(\nu)}), \qquad \text{where } a_{j,e}^{*(\nu)} = a_\nu + r_e^* \nu_{j,e}^*, \;\; b_{j,e}^{*(\nu)} = b_\nu + 1 - r_e^* \nu_{j,e}^*.$$

$$q(\rho_{v,e}) = \text{Beta}(a_{v,e}^{*(\rho)}, b_{v,e}^{*(\rho)}), \qquad \text{where } a_{v,e}^{*(\rho)} = a_\rho + r_e^* \rho_{v,e}^*, \;\; b_{v,e}^{*(\rho)} = b_\rho + 1 - r_e^* \rho_{v,e}^*.$$

$$q(r_e) = \text{Beta}(a_e^{*(r)}, b_e^{*(r)}), \qquad \text{where } a_e^{*(r)} = a_r + r_e^*, \;\; b_e^{*(r)} = b_r + 1 - r_e^*.$$

A.4.2. Expectations

$$\mathbb{E}_q \left[ \log \rho_{v,e} \right] = \psi \left( a_{v,e}^{*(\rho)} \right) - \psi \left( a_{v,e}^{*(\rho)} + b_{v,e}^{*(\rho)} \right), \quad \mathbb{E}_q \left[ \log(1 - \rho_{v,e}) \right] = \psi \left( b_{v,e}^{*(\rho)} \right) - \psi \left( a_{v,e}^{*(\rho)} + b_{v,e}^{*(\rho)} \right)$$

$$\mathbb{E}_q \left[ \log \nu_{j,e} \right] = \psi \left( a_{j,e}^{*(\nu)} \right) - \psi \left( a_{j,e}^{*(\nu)} + b_{j,e}^{*(\nu)} \right), \quad \mathbb{E}_q \left[ \log(1 - \nu_{j,e}) \right] = \psi \left( b_{j,e}^{*(\nu)} \right) - \psi \left( a_{j,e}^{*(\nu)} + b_{j,e}^{*(\nu)} \right)$$

$$\mathbb{E}_q \left[ \log r_e \right] = \psi \left( a_e^{*(r)} \right) - \psi \left( a_e^{*(r)} + b_e^{*(r)} \right), \quad \mathbb{E}_q \left[ \log(1 - r_e) \right] = \psi \left( b_e^{*(r)} \right) - \psi \left( a_e^{*(r)} + b_e^{*(r)} \right)$$

$$\mathbb{E}_q \left[ \sigma_\mu^{-2} \right] = \frac{a_\mu^*}{b_\mu^*}, \quad \mathbb{E}_q \left[ \log \sigma_\mu^2 \right] = \log b_\mu^* - \psi(a_\mu^*), \quad \mathbb{E}_q \left[ \omega_{i,v} \right] = \frac{1}{2\eta_{i,v}^*} \tanh \left( \eta_{i,v}^*/2 \right)$$

$$\mathbb{E}_q \left[ O(S_e, S_{e'}) \right] \approx \frac{\left[ \boldsymbol{\rho}_{\cdot,e}^* \right]^\top \boldsymbol{\rho}_{\cdot,e'}^*}{\min \left( \sum_{v=1}^{V} \rho_{v,e}^*, \sum_{v=1}^{V} \rho_{v,e'}^* \right)}$$

$$\boldsymbol{h}_e^{(v=1)} = \left( \rho_{1,e}^*, \rho_{2,e}^*, \ldots, \rho_{v-1,e}^*, 1, \rho_{v+1,e}^*, \ldots, \rho_{V,e}^* \right)^\top, \;\; \boldsymbol{h}_e^{(v=0)} = \left( \rho_{1,e}^*, \rho_{2,e}^*, \ldots, \rho_{v-1,e}^*, 0, \rho_{v+1,e}^*, \ldots, \rho_{V,e}^* \right)^\top,$$

$$\mathbb{E}_q \left[ O \left( S_e, S_{e'} \mid m_{v,e} = 1 \right) - O \left( S_e, S_{e'} \mid m_{v,e} = 0 \right) \right] = \frac{\left[ \boldsymbol{h}_e^{(v=1)} \right]^\top \boldsymbol{\rho}_{\cdot,e'}^*}{\min \left( \| \boldsymbol{h}_e^{(v=1)} \|_1, \sum_{v=1}^{V} \rho_{v,e'}^* \right)} - \frac{\left[ \boldsymbol{h}_e^{(v=0)} \right]^\top \boldsymbol{\rho}_{\cdot,e'}^*}{\min \left( \| \boldsymbol{h}_e^{(v=0)} \|_1, \sum_{v=1}^{V} \rho_{v,e'}^* \right)}$$

$$\mathbb{E}_q\left[\eta_{i,v}\right] = d_v^{-1}\sum_{e=1}^{E} r_e^* \rho_{v,e}^* \left[\sum_{j=1}^{P} x_{i,j}\nu_{j,e}^*\mu_{j,e}^*\right], \quad \mathbb{E}_q\left[\tilde{\eta}_{i,v}\right] = \mathbb{E}_q\left[\eta_{i,v}\right] + \alpha_v^*$$

$$\mathbb{E}_q\left[b_{i,e}^2 \mid z_e = 1\right] = \sum_{j=1}^{P} x_{i,j}^2 \nu_{j,e}^* \left[\left(\mu_{j,e}^*\right)^2 + \sigma_{j,e}^{2*}\right] + 2\sum_{j<k} x_{i,j}x_{i,k}\nu_{j,e}^*\mu_{j,e}^*\nu_{k,e}^*\mu_{k,e}^*$$

$$\mathbb{E}_q\left[\eta_{i,v}^2\right] = d_v^{-2}\sum_{e} r_e^* \rho_{v,e}^* \mathbb{E}_q\left[b_{i,e}^2 \mid z_e = 1\right] + 2d_v^{-2}\sum_{e<e'} r_e^* r_{e'}^* \rho_{v,e}^* \rho_{v,e'}^* \left[\sum_{j=1}^{P} x_{i,j}\nu_{j,e}^*\mu_{j,e}^*\right]\left[\sum_{j=1}^{P} x_{i,j}\nu_{j,e'}^*\mu_{j,e'}^*\right]$$

$$\mathbb{E}_q\left[\tilde{\eta}_{i,v}^2\right] = \mathbb{E}_q\left[\eta_{i,v}^2\right] + \left[\left(\alpha_v^*\right)^2 + \varrho_v^{2*}\right] + 2\alpha_v^* \mathbb{E}_q\left[\eta_{i,v}\right]$$

$$\mathbb{E}_q\left[\xi_{v,e} \mid z_e = 1\right] = \sum_{i=1}^{N} R_{i,v}\mathbb{E}_q\left[\omega_{i,v}\right]\mathbb{E}_q\left[b_{i,e}^2 \mid z_e = 1\right]$$

$$\mathbb{E}_q\left[a_{i,v}^{(-e)}\right] = \mathbb{E}_q\left[\eta_{i,v}\right] - d_v^{-1} r_e^* \rho_{v,e}^* \left[\sum_{j=1}^{P} x_{i,j}\nu_{j,e}^*\mu_{j,e}^*\right]$$

$$\mathbb{E}_q\left[a_{i,v}^{\backslash(j,e)} \mid z_e = 1\right] = \mathbb{E}_q\left[a_{i,v}^{(-e)}\right] + d_v^{-1}\rho_{v,e}^* \left[\sum_{j=1}^{P} x_{i,j}\nu_{j,e}^*\mu_{j,e}^*\right] - d_v^{-1}\rho_{v,e}^* x_{i,j}\nu_{j,e}^*\mu_{j,e}^*$$

$$\mathbb{E}_q\left[\zeta_{v,e} \mid z_e = 1\right] = \sum_{i=1}^{N} R_{i,v}\left(\kappa_{i,v} - \mathbb{E}_q\left[\omega_{i,v}\right]\left\{\mathbb{E}_q\left[a_{i,v}^{(-e)}\right] + \alpha_v^*\right\}\right)\left[\sum_{j=1}^{P} x_{i,j}\nu_{j,e}^*\mu_{j,e}^*\right]$$

## B. Additional Simulation Results

### B.1. Visualization of Structure Discovery

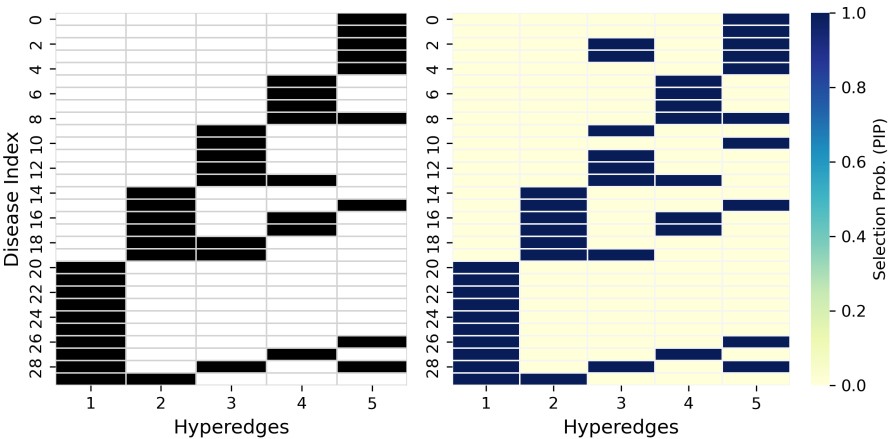

*Figure 6.* Recovery of latent disease hypergraph. Heatmaps show true (left) and inferred (right) disease hypergraph incidence matrices for a representative simulation replicate. BHPI accurately recovers overlapping hypergraph structure with limited redundancy.

To qualitatively assess the structural recovery performance of BHPI, we visualize the latent disease hypergraph incidence matrix $H$ and the risk-factor-hyperedge effect selector $\boldsymbol{\gamma}_j$, along with the effect size $\boldsymbol{\mu}_j$. Figure 6 compares the ground-truth incidence matrix against the posterior mean inferred by BHPI. The model successfully identifies the block-diagonal structure and handles overlapping memberships without introducing significant noise. Furthermore, Figure 7 illustrates that BHPI accurately captures the sparsity and magnitude of feature effects, correctly assigning risk factors to their respective latent pathways.

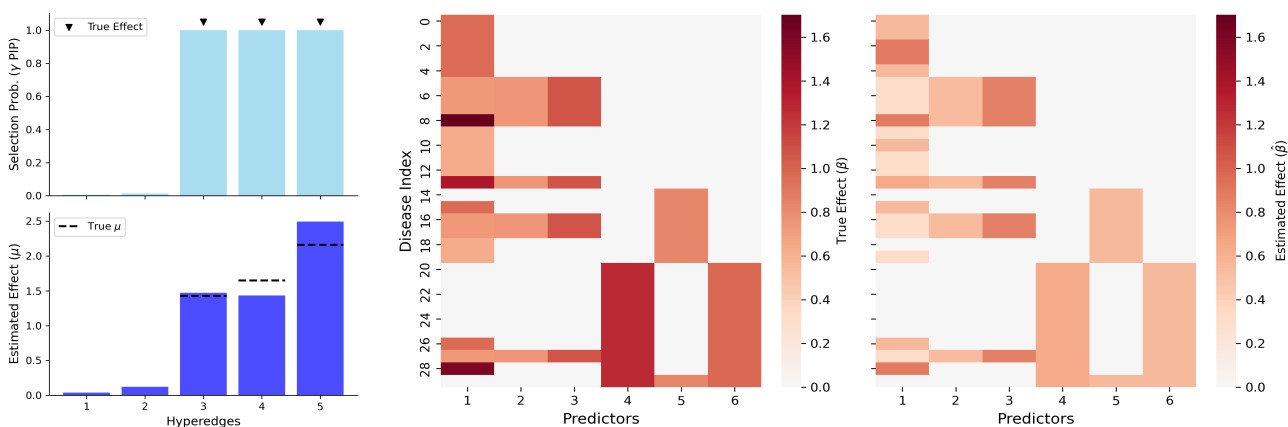

*Figure 7.* Bar plots show true versus inferred effects of predictor 1 across hyperedges, measured by PIP of $\gamma$ and magnitude $\mu$.

## B.2. Ablation: Role of the Repulsion Prior

*Table 7.* Role of the repulsion prior in hypergraph identifiability ($N = 2000$ and $5000$). Predictive performance remains stable, while repulsion reduces redundancy and improves stability of inferred hyperedges, as measured by Jaccard similarity (Jacc.), overlap coefficients (Ov.) across replicates, effective hyperedge (Eff.#/RF) per risk factor (RF), and hyperedge overlap per RF (Ov./RF).

| | $N = 2000$ | | | | | $N = 5000$ | | | | |
| $\lambda$ | AUC | JACC. | OV. | EFF.#/RF | OV./RF | AUC | JACC. | OV. | EFF.#/RF | OV./RF |
|---|---|---|---|---|---|---|---|---|---|---|
| $= 0$ | 74.69 | 0.542 | 0.800 | 1.992 | 0.097 | 74.51 | 0.581 | 0.836 | 2.392 | 0.116 |
| $> 0$ | 75.00 | 0.565 | 0.812 | 1.874 | 0.082 | 74.63 | 0.619 | 0.830 | 2.151 | 0.102 |

The repulsion prior $\mathcal{R}_{\text{rep}}$ is designed to enforce identifiability by penalizing redundant latent pathways. In this section, we expand on the results in Table 3, as displayed in Table 7. While predictive AUC is relatively insensitive to the repulsion strength $\lambda$, the structural metrics, specifically Jaccard similarity, show marked improvement when $\lambda > 0$. Without repulsion, the model often instantiates multiple near-identical hyperedges to represent a single disease pathway, leading to a "collapsed" latent space that is difficult to interpret.

## B.3. Sensitivity Analysis

We evaluate the robustness of BHPI to the repulsion strength $\lambda$, and the assumed maximum number of latent hyperedges $E$. Regarding $\lambda$, as shown in Figure 8, we observe a wide "stable regime" where the balance between sparsity and structure recovery remains optimal, suggesting that BHPI does not require extensive hyperparameter tuning to achieve reliable results. Moreover, from Figure 9, we find that increasing $\lambda$ sharpens posterior inclusion probabilities (PIPs) and reduces redundancy between inferred hyperedges, leading to more stable latent structures across random splits. We further examine robustness to over-specification of model capacity by varying $E$ as displayed in Figure 10, and find that as $E$ increases beyond the "true" number of pathways, the variational factor $q(z_e)$ effectively prunes the excess components. This shows that even when $E$ is over-specified, the effective number of active hyperedges remains stable, demonstrating effective self-regularization.

## B.4. Structural-PIP Calibration: CAVI vs MCMC

To validate the accuracy of the structural posterior, we compute PIP expected calibration error (ECE) for $H$, $z$, and $\gamma$ on the paper's simulation setting and compare against a full Gibbs sampler with Pólya–Gamma augmentation ($\mu = 1.5$, $\sigma = 0.5$, 50 replications; Table 8). These results show that CAVI and MCMC achieve comparable PIP calibration on the structural variables (ECE 0.03–0.09 for both methods), with MCMC performing slightly better. Structural recovery and predictive AUC are also closely matched. Taken together, these diagnostics support the more precise statement that BHPI provides well-supported structural posterior uncertainty and competitive predictive calibration.

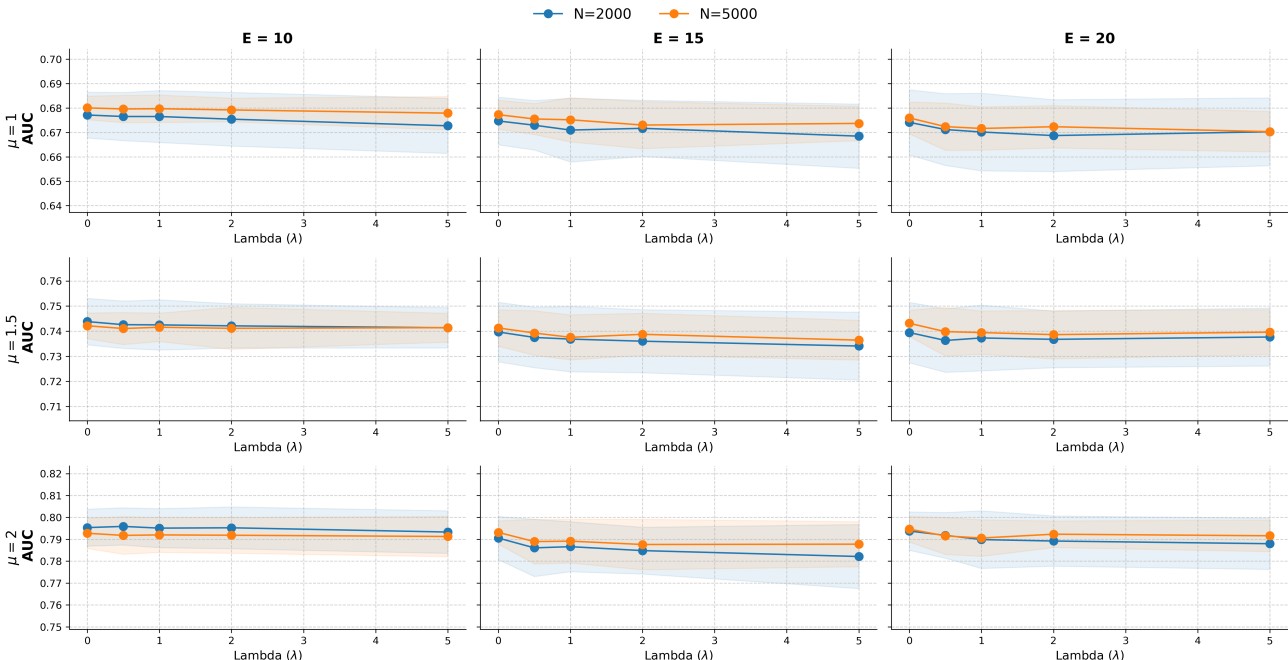

*Figure 8.* Robustness of predictive performance to repulsion strength across model sizes. Each panel fixes the number of hyperedges $E$; predictive AUC remains stable across a wide range of $\lambda$ and $E$, indicating robustness to both hyperparameters.

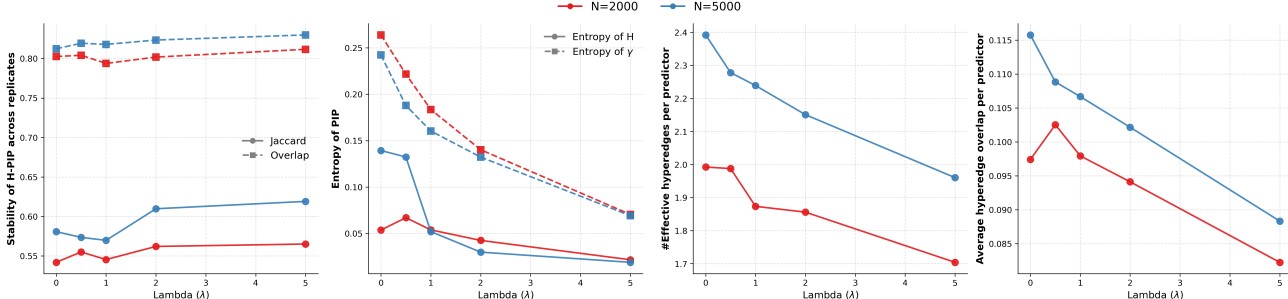

*Figure 9.* Structural stability, sharpness, and redundancy across repulsion strengths.

### B.5. Initialization Stability and Identifiability

Empirically, after permutation alignment, the hypergraph overlap across different initializations on the same data is $0.905 \pm 0.062$ in simulation and $0.875 \pm 0.044$ on UKB. Cross-replicate overlap in simulation is $0.81 - 0.83$. Predictive variability remains small throughout (AUC std 0.01). In practice, we run CAVI from multiple initializations and select the model with the highest validation performance. Among the selected models, the core hypergraph structure — which edges are active and which diseases are confidently assigned — is consistent; the remaining differences involve borderline diseases whose membership PIPs are near 0.5. Exact hyperedge-level identifiability is not guaranteed: the repulsion prior reduces redundancy but does not break permutation symmetry of the latent hyperedges. For this reason, our scientific claims do not rely on a unique labeled decomposition, but instead on quantities that are stable under relabeling: permutation-aligned overlap, pairwise co-membership structure, disease-level effects, and held-out predictions.

## C. Additional Results for UKB Application

In this section, we provide additional information and results for the UK Biobank (UKB) application. This includes detailed dataset characteristics in Table 9, receiver operating characteristic (ROC) analysis for several representative diseases, and an expanded library of qualitative case studies that illustrate the clinical interpretability of the learned latent hypergraph.

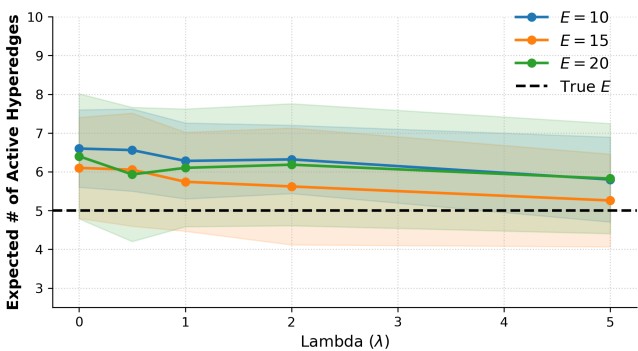

*Figure 10.* Robustness to over-specification of the maximum number of hyperedges $E$, measured by the posterior expected number of active hyperedges.

*Table 8.* Structural-PIP calibration: CAVI vs full Gibbs MCMC sampler (mean $\pm$ s.d. across 50 replicates). ECE is the per-variable expected calibration error of the PIPs.

| Method | $H$ overlap | $\text{ECE}_H$ | $\text{ECE}_z$ | $\text{ECE}_\gamma$ | Pred. AUC |
|---|---|---|---|---|---|
| MCMC | $0.885 \pm 0.068$ | $0.081 \pm 0.019$ | $0.058 \pm 0.053$ | $0.039 \pm 0.032$ | $0.747 \pm 0.009$ |
| CAVI | $0.850 \pm 0.054$ | $0.087 \pm 0.026$ | $0.028 \pm 0.081$ | $0.050 \pm 0.030$ | $0.741 \pm 0.008$ |

The cohort characteristics, summarized in Table 9, reflect the high-dimensional and imbalanced nature of real-world clinical data.

*Table 9.* UK Biobank dataset characteristics. The cohort exhibits high-dimensional predictors, substantial class imbalance, and a mix of common and rare disease outcomes.

| STATISTIC | VALUE |
|---|---|
| PARTICIPANTS ($N$) | 277,291 |
| RISK FACTORS ($P$) | 71 |
| DISEASE OUTCOMES ($V$) | 66 |
| *Disease Prevalence (%)* | |
|   MEDIAN | 1.8% |
|   RANGE (MIN – MAX) | 1.0% – 21.8% |
| *Data Sparsity* | |
|   AVG. DISEASES PER PATIENT | 4.2 |
|   OBSERVATION WINDOW | 19 YEARS |

## C.1. ROC for Representative Diseases

To complement the aggregate performance metrics reported in the main text, we visualize the predictive stability of BHPI across diseases with varying prevalence. As shown in Figure 11, BHPI maintains competitive AUCs even for low-prevalence conditions (e.g., prevalence $\approx 1\%$), where baseline models often exhibit higher variance or sensitivity to class imbalance.

## C.2. Extended Qualitative Case Studies

While the main text highlights the "Smoking" and "Dementia" examples, to further validate the clinical utility of BHPI, we provide an extensive analysis of additional latent pathways recovered from the UK Biobank data. These cases illustrate the model's ability to handle complex geriatric, metabolic, and psychiatric comorbidities.

### C.2.1. DIVERGENT RISK PATHWAYS

**Case A: The Systemic Impact of Sleeplessness.** We examine "Sleeplessness / Insomnia," a risk factor often treated as a generic marker of poor health. BHPI reveals that this feature actively modulates distinct phenotypic pathways: **(i)**

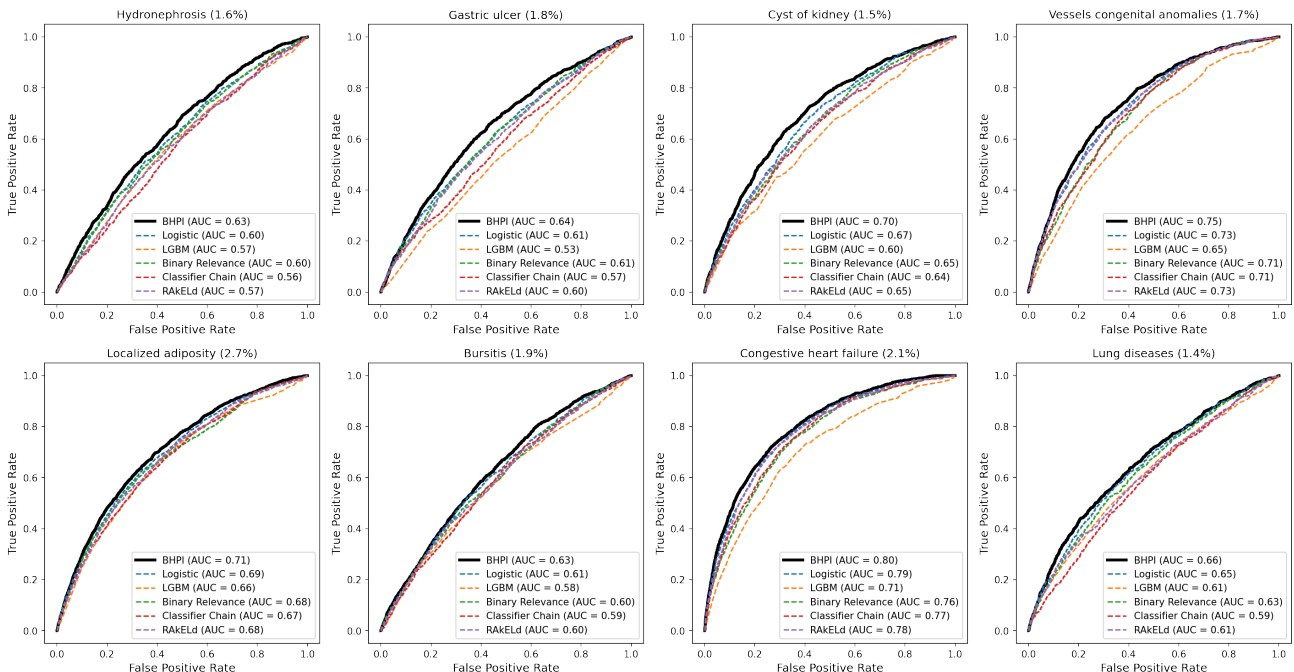

*Figure 11.* ROC curves of representative diseases for CAVI, logistic regression, and LGBM, with the prevalence indicated in brackets.

**The Mental Health Pathway (Hyperedge 22):** Sleeplessness is a primary driver of a highly specific cluster ($V = 1$) containing only *Other mental disorder*. Modulated by *Sex* and *Chest pain*, this captures the bidirectional link between sleep and psychiatric instability. **(ii) The Cardio-Metabolic Pathway (Hyperedge 16):** In contrast, Sleeplessness also drives Hyperedge 16 ($V = 17$), co-occurring with *Obesity*, *Type 2 Diabetes*, and *Sleep apnea*. Here, it interacts with lifestyle predictors like *Diet* and *BMI*, reflecting the physiological impact of sleep deprivation on metabolic regulation. **(iii) The Genitourinary Pathway (Hyperedge 27):** Finally, it links to *Hyperplasia of prostate*, capturing the symptom-driven association where nocturia disrupts sleep.

**Case B: Alcohol: Organ Toxicity vs. Caloric Impact.** **(i) The Liver-Toxicity Pathway (Hyperedge 18):** Alcohol intake predicts Hyperedge 18 ($V = 11$), which includes *Nonalcoholic liver disease* and *Cholelithiasis*. This isolates the direct toxic/inflammatory effects on the hepatic system. **(ii) The Pure Obesity Pathway (Hyperedge 25):** Distinct from organ damage, Alcohol intake also drives Hyperedge 25 ($V = 1$), which contains only *Obesity*. This isolates the caloric contribution of alcohol to body mass separate from pathophysiological organ damage.

C.2.2. CONVERGENT DISEASE PATHWAYS

**Case C: Type 2 Diabetes Contexts.** **(i) Active Metabolic Syndrome (Hyperedge 16):** Diabetes appears with *Obesity* and *Sleep apnea*, modulated by modifiable lifestyle factors (Diet, Physical activity). This represents the active, manageable phase of the disease. **(ii) Cardiovascular End-Point (Hyperedge 4):** Diabetes also appears in Hyperedge 4 ($V = 29$), dominated by *Heart failure* and *Atherosclerosis*. Here, predictors are markers of established damage (*Chest pain*, *Wheeze*), modeling Diabetes as a contributor to terminal cardiovascular collapse.

**Case D: Obesity Heterogeneity.** **(i) Frailty & Structural Pathway (Hyperedge 20):** Obesity co-occurs with *Prolapse of vaginal walls* and *Osteoporosis*, modulated by *Age* and *Sex*. **(ii) Metabolic-Digestive Pathway (Hyperedge 13):** Obesity is grouped with *Type 2 Diabetes* and *Liver disease*, driven by *Employment* and *Walking frequency*.

# D. Computational Cost

### D.1. Theoretical Complexity

As discussed in Section 4.2, the per-iteration computational complexity of the BHPI coordinate ascent algorithm is approximately $\mathcal{O}(N \cdot E \cdot (P + V))$. This linear scaling with respect to the number of samples ($N$), the number of risk factors ($P$), and the number of diseases ($V$) is a critical property that enables the framework to handle biobank-scale data. The primary computational bottleneck lies in the simultaneous update of the hyperedge existence $z_e$, the disease-membership indicators $m_{v,e}$ and the feature-effect selectors $\gamma_{j,e}$, all of which require aggregating information across all $N$ individuals in the dataset. However, because the variational updates for different hyperedges $e \in \{1, \ldots, E\}$ are independent within each iteration, the algorithm can be trivially parallelized to further reduce wall-clock time in high-performance computing environments.

### D.2. Empirical Scaling with Model Capacity

To evaluate how the "structural discovery tax" grows with model complexity, we measured the training time on the simulated dataset while varying the maximum number of latent hyperedges $E$ ($E = 10, 15$ and $20$ respectively). As shown in the scaling results in Table 10, the training time grows linearly with $E$. This confirms that our repulsion-aware inference remains stable even as model capacity increases, as the variational "global switch" $r_e^*$ efficiently prunes redundant components without incurring exponential computational overhead.

While the proposed BHPI model incurs a higher computational cost during the training phase compared to independent baseline models, it remains highly scalable and practical for biobank-scale datasets. As summarized in Table 10, the training time for nearly 300,000 samples is approximately 74 minutes, a manageable one-time "structural discovery tax" for learning a complex, interpretable latent hypergraph. Crucially, while BHPI requires more time for one-time structural discovery, its inference latency remains on par with simple logistic regression ($< 0.1$ ms), ensuring the model's suitability for real-time clinical decision support even at biobank scale. While BHPI requires more memory (27.8 GB) to maintain its structured variational parameters, this remains well within the capacity of standard modern research servers, proving that the model can effectively scale to the high-dimensional requirements of modern EHR analysis without compromising on predictive speed.

*Table 10.* Wall-clock runtimes for simulated data and UK Biobank application ($N \approx 277,291$; $V = 66$). Synthetic tests were conducted on a single-core Intel Xeon 6342 (4 GB RAM); UK Biobank results utilized 4 cores and 60 GB RAM. BHPI maintains inference latency on par with Logistic Regression ($< 0.1$ ms) while providing structured latent discovery.

| | Simulated Data (Section 5.1) | | UK Biobank | | | |
|---|---|---|---|---|---|---|
| **Model** | **Runtime (sec)** $N = 2000$ | **Runtime (sec)** $N = 5000$ | **Model** | **Memory (GB)** | **Training (min)** | **Inference (ms)** |
| Logistic Reg. | 0.12 | 0.23 | Logistic Reg. | 3.74 | 5.82 | $< 0.1$ |
| LightGBM | 3.56 | 6.14 | LightGBM | 3.18 | 17.41 | 0.48 |
| Binary Relevance | 0.30 | 0.61 | Binary Relevance | 1.45 | 12.24 | 0.29 |
| Classifier Chains | 0.33 | 0.68 | Classifier Chains | 1.36 | 8.23 | 0.56 |
| RAkELd | 0.59 | 1.35 | RAkELd | 1.39 | 3.06 | 1.56 |
| **BHPI** ($E = 10/15/20$) | **1.6/2.7/3.2** | **4.1/6.4/9.3** | **BHPI** ($E = 60$) | **27.8** | **73.7** | $< 0.1$ |

