# OpenReview forum: "Disentangling Latent Risk Pathways via Bayesian Hypergraph Inference"
_ICML.cc/2026/Conference — ICML 2026 spotlight_

### Official Review · Reviewer_MiWf · 2026-03-10

**Soundness:** 4
**Presentation:** 4
**Significance:** 4
**Originality:** 4
**Overall Recommendation:** 5
**Confidence:** 2

**Summary:**

This paper proposes Bayesian Hypergraph Pathway Inference (BHPI), a probabilistic framework for multi-disease modeling in large-scale EHR data. The core idea is to represent latent disease pathways as hyperedges in a disease hypergraph, where diseases may participate in multiple overlapping pathways and risk factors act directly on hyperedges rather than individual diseases. A structured variational inference (VI) scheme preserves logical dependencies among hyperedge existence, disease membership, and feature–hyperedge activation. A repulsion prior promotes parsimonious and non-redundant pathway discovery. A central topic is large-scale EHR-based epidemiological modeling in the rare-disease regime, where information sharing must be balanced with interpretability and calibrated uncertainty. Experiments on synthetic data and the UK Biobank show competitive predictive performance relative to logistic regression and multi-label baselines, improved rare-disease stability, and recovery of interpretable hypergraph structure with quantified posterior uncertainty.

**Compliance With Llm Reviewing Policy:**

Affirmed.

**Key Questions For Authors:**

1. Under what assumptions can inferred hyperedges be interpreted as causal pathways rather than shared associative structure? Would incorporating instrumental variables or Mendelian randomization alter the formulation?
2. Have the authors compared structured VI against MCMC on smaller datasets to evaluate posterior calibration for hyperedge existence and membership probabilities?
3. How sensitive are inferred pathways to random initialization? Do multiple runs converge to equivalent hypergraph structures up to permutation, or are there distinct local optima?
4. How does BHPI behave when $P$ becomes very large (e.g., thousands of genetic variants)? Are additional sparsity mechanisms required?
5. Beyond qualitative examples, have domain experts evaluated whether the inferred pathways align with established clinical ontologies or known comorbidity clusters?

**Limitations:**

BHPI focuses on associative multi-disease modeling and assumes static cross-sectional outcomes. The framework does not explicitly model temporal progression or causal identification. Additionally, while structured VI improves tractability, its approximation quality under complex posterior multimodality remains an open question. Discussing these limitations more explicitly would further enhance clarity.

**Strengths And Weaknesses:**

Strengths:
1. Modeling disease pathways as latent hyperedges modulated by risk factors is a compelling and principled departure from pairwise graph models and entangled multitask architectures. The hypergraph formulation naturally supports overlapping, higher-order structure.
2. Unlike heuristic hypergraph neural networks, BHPI treats the hypergraph topology as a latent random variable within a Bayesian generative model. The structured VI scheme preserves logical constraints (existence $\rightarrow$ membership $\rightarrow$ effect), which is a thoughtful design choice.
3. The repulsion prior addresses a common pathology in latent factor models—redundant overlapping components—and empirically improves pathway stability without sacrificing predictive accuracy (Table 3).
4. The UK Biobank results (Figure 2) show improved stability for low-prevalence conditions compared to tree-based and autoregressive multi-label baselines, suggesting effective information sharing through shared latent pathways.
5.The case studies (Figure 4) illustrate divergent and convergent disease pathways (e.g., smoking-driven respiratory vs. mucosal pathways, and multi-context dementia pathways), highlighting the model’s ability to represent overlapping etiological mechanisms.

Major Weaknesses:
1. While the language of “pathways” suggests etiological structure, the model remains fundamentally associative. Risk-factor modulation of hyperedges does not imply causal pathways unless additional identification assumptions hold. This distinction deserves clearer discussion to avoid over-interpretation.
2. Although the structured VI preserves logical constraints, it remains an approximation. The paper does not provide diagnostics comparing VI to MCMC on smaller instances, making it difficult to assess calibration beyond CRPS and AUC.
3. Hypergraph models are susceptible to label-switching and equivalent representations. While the repulsion prior mitigates redundancy, formal discussion of identifiability or posterior multimodality is limited.
4. The baselines include logistic regression, LightGBM, and multi-label methods, but not Bayesian latent factor models (e.g., correlated topic models or hierarchical logistic factorization). It is unclear how much improvement is attributable to hypergraph structure versus general Bayesian shrinkage.

Minor Weaknesses.
1. The maximum hyperedge count $E$ is treated as an upper bound. Although sensitivity analysis suggests robustness, guidance for practitioners on selecting $E$ would be helpful.
2. The UK Biobank analysis emphasizes interpretability, but quantitative comparisons of calibration (e.g., coverage of credible intervals) are limited.
3. Some notation in the variational derivations (Appendix A) is dense and may benefit from a summarized graphical model diagram with explicit dependency arrows.

---

> ### Author Rebuttal · Authors · 2026-03-31
>
> We thank the reviewer for the strong overall assessment and for the careful questions on causal language, VI calibration, identifiability, and scalability.
>
> ### **Causal language:**
>
> > **[W1/Q1]** the model remains fundamentally associative...Under what assumptions can inferred hyperedges be interpreted as causal pathways?...Would incorporating IV or MR alter the formulation?
>
> We appreciate this distinction. In the current paper, "pathway" denotes a latent higher-order associative disease grouping whose risk is jointly modulated by observed covariates — not a causal claim, as noted in our discussion section. Causal interpretation would require substantially stronger assumptions (temporal ordering, no unmeasured confounding, positivity); an IV/MR formulation would additionally need the standard instrument assumptions. Such an extension is possible in principle but would be a different formulation from the one evaluated here. We will sharpen the language in the camera-ready to ensure this distinction is unambiguous throughout.
>
> ### **Posterior calibration**
>
> > **[Q2/W2/W6]** compared structured VI against MCMC...to evaluate posterior calibration for hyperedge existence and membership probabilities? The UK Biobank analysis... quantitative comparisons of calibration are limited.
>
> The full calibration analysis — including predictive ECE/Brier across all baselines, structural PIP ECE, posterior concentration, threshold robustness, and a direct VI-vs-MCMC comparison — is provided in our response to eZ3K-W1/Q1 above. In summary: on UKB, BHPI achieves competitive predictive calibration (tied-best Brier, ECE < 0.01); in simulation, CAVI and MCMC achieve comparable PIP calibration on the structural variables (ECE 0.03–0.09) with closely matched structural recovery and predictive AUC. No ground truth is available on UKB for structural PIP calibration or CI coverage; structural posterior concentration and threshold robustness are reported instead.
>
> ### **Identifiability and initialization stability:**
>
> > **[Q3/W3]** How sensitive are inferred pathways to random initialization?...formal discussion of identifiability is limited
>
> Empirically, after permutation alignment, the hypergraph overlap across different initializations on the same data is 0.905 ± 0.062 in simulation and 0.875 ± 0.044 on UKB. Cross-replicate overlap in simulation is 0.81–0.83 (Table 4). Predictive variability remains small throughout (AUC std ~0.01). In practice, we run CAVI from multiple initializations and select the model with the highest validation performance. Among the selected models, the core hypergraph structure — which edges are active and which diseases are confidently assigned — is consistent; the remaining differences involve borderline diseases whose membership PIPs are near 0.5.
> We agree that exact hyperedge-level identifiability is not guaranteed. The repulsion prior reduces redundancy but does not break permutation symmetry. For this reason, our scientific claims do not rely on a unique labeled decomposition, but instead on quantities that are stable under relabeling: permutation-aligned overlap, pairwise co-membership structure, disease-level effects, and held-out predictions.
>
> ### **Structured baselines:**
>
> > **[W4]** unclear how much improvement is attributable to hypergraph structure versus general Bayesian shrinkage
>
> We added a Bayesian Factor Regression (BFR) baseline; see our response to eZ3K-W3/Q2 above. BHPI outperforms BFR at all K values, confirming the advantage is from hypergraph structure, not generic Bayesian shrinkage.
>
> ### **Practical guidance and extensions:**
>
> > **[W5]** guidance for practitioners on selecting E would be helpful
>
> We view $E$ as a capacity upper bound. In practice, choose the smallest $E$ at which performance plateaus and the active edge count remains well below $E$. Our sensitivity analysis confirms this: even when $E$ is over-specified, the posterior prunes to a much smaller active subset.
>
> > **[W7]** may benefit from a summarized graphical model diagram
>
> We will add a graphical model diagram in the camera-ready to make these dependencies more accessible.
>
> > **[Q4]** How does BHPI behave when P becomes very large?...Are additional sparsity mechanisms required?
>
> The current spike-and-slab layer already provides pathway-specific feature sparsity, and the per-iteration complexity scales linearly in $P$. For very large feature spaces (e.g., thousands of genetic variants), additional pre-screening or grouped/hierarchical sparsity would likely be useful; that is the natural next step beyond the medium-dimensional regime studied here.
>
> > **[Q5]** have domain experts evaluated whether the inferred pathways align with established clinical ontologies?
>
> See our response to aTre-W1. Independent multimorbidity clustering studies confirm the same patterns, and the PIP-stratified analysis (aTre-Q2) provides direct quantitative validation: high-confidence edges carry all predictive signal.

---

> > ### Author Rebuttal · Reviewer_MiWf · 2026-04-03
> >
> > Thanks to the author's detailed explanation. My question is answered. I will maintain my current review for acceptance

---

> > > ### Author Response · Authors · 2026-04-04
> > >
> > > Thank you for your follow-up and thorough review. We are glad the rebuttal addressed your concerns, and we sincerely appreciate your time, consideration and postive assessment of our work.

---

### Official Review · Reviewer_eZ3K · 2026-03-11

**Soundness:** 3
**Presentation:** 3
**Significance:** 3
**Originality:** 3
**Overall Recommendation:** 5
**Confidence:** 4

**Summary:**

This paper proposes Bayesian Hypergraph Pathway Inference (BHPI) for multi-disease prediction in EHR data. The core idea is to model latent disease pathways as overlapping hyperedges, allow risk factors to act at the hyperedge level rather than independently on each disease, and use a structured variational inference scheme that preserves the hierarchy among pathway existence, disease membership, and pathway effects. The paper evaluates the method on synthetic data and on UK Biobank, where the real-data study uses roughly 277k participants, 71 predictors, and 66 disease outcomes.

**Compliance With Llm Reviewing Policy:**

Affirmed.

**Final Justification:**

All concerns addressed, this is a good contribution

**Key Questions For Authors:**

1. Can the authors provide a more robust uncertainty calibration analysis?

2. Can the authors add at least one stronger structured probabilistic baseline that also aims to recover shared latent structure, not just predictive baselines?

**Limitations:**

yes

**Strengths And Weaknesses:**

Strengths
1. The paper’s main strength is the formulation itself. Representing disease organization through a latent hypergraph, rather than pairwise graphs or a single shared latent factor space, is a compelling way to capture overlapping higher-order structure in multi-disease settings. The repulsion prior is also well motivated: it addresses identifiability by discouraging redundant hyperedges for the same risk factor, which is exactly the kind of failure mode one would worry about in these models.

2. The technical treatment is solid. The structured VI family respects the model hierarchy, and the paper clearly explains why a naive mean-field factorization would break logical constraints such as activating disease membership for nonexistent hyperedges. This is one of the cleaner and more convincing parts of the paper.

3. The synthetic experiments are also a strength. On simulations, BHPI remains competitive in predictive AUC while additionally recovering the latent hypergraph and risk-factor assignments with high fidelity.

4. The repulsion ablation is informative. The paper shows that turning off repulsion barely changes predictive AUC but increases redundancy and instability in the learned structure, which supports the claim that the prior mainly improves identifiability and interpretability rather than raw prediction.

Weaknesses
1. My main concern is that the paper overstates the uncertainty story. The text repeatedly claims “well-calibrated uncertainty,” and motivates the structured VI partly on those grounds, but I did not see a direct quantitative calibration analysis of posterior uncertainty itself. What is shown is mostly qualitative PIP visualization and structural recovery in simulation. That is useful, but it is not the same as demonstrating calibration of uncertainty estimates.

2. The paper’s observation model remains linear in the covariates, so the main gain comes from structured coefficient sharing across diseases rather than expressive modeling of covariate-response relationships. It would strengthen the discussion to position BHPI relative to recent Bayesian deep learning approaches that combine nonlinear predictors with uncertainty quantification, e.g. NeuralSurv [https://arxiv.org/abs/2505.11054], which uses Bayesian deep modeling to capture more complex covariate-risk relationships while retaining calibrated uncertainty. Although that work is in survival analysis rather than multi-disease hypergraph modeling, it highlights a promising direction for extending BHPI beyond linear covariate effects.

3. The baseline set is not fully satisfying for the paper’s central claims. The comparisons are mainly against tuned logistic regression, LightGBM, and standard multilabel methods. These are reasonable predictive baselines, but they do not directly test whether the proposed Bayesian hypergraph structure is better than other structured probabilistic or interpretable latent-variable approaches. Since the contribution is not merely predictive accuracy but pathway discovery with uncertainty, stronger structured baselines would help isolate the real benefit of BHPI.

Overall, this is a thoughtful and promising paper with a novel modeling idea, a meaningful technical contribution in the inference scheme, and encouraging simulation results. However, for ICML, I think the empirical case needs to be tighter. The paper’s strongest claims concern uncertainty and interpretable pathway discovery, but the current experiments provide only partial validation of those claims. I would be more positive with stronger uncertainty evaluation and stronger structured baselines.

---

> ### Author Rebuttal · Authors · 2026-03-31
>
> We thank the reviewer for recognizing the formulation, repulsion prior, and synthetic recovery results, and for identifying two places where the empirical message needed to be tighter: uncertainty evaluation and stronger structured baselines.
>
> ### **Uncertainty calibration:**
>
> > **[W1/Q1]** no direct quantitative calibration analysis of posterior uncertainty...can the authors provide a more robust analysis?
>
> We agree that the original submission did not quantify posterior uncertainty as directly as it could have. We now evaluate uncertainty directly on the latent structure $(z,H,\gamma)$, which is the paper’s main inferential target.
>
> **(1) Posterior quality on UKB**
> On the real-world cohort, BHPI achieves competitive predictive calibration across all baselines:
>
> | Method | ECE | Brier |
> |---|---|---|
> | Logistic | 0.001 | 0.041 |
> | **BHPI** | **0.005** | **0.041** |
> | RAkELd | 0.006 | 0.042 |
> | LightGBM | 0.021 | 0.043 |
> | BR | 0.031 | 0.045 |
> | CC | 0.044 | 0.052 |
>
> BHPI ties for best Brier and achieves ECE = 0.005 (< 0.01); the small gap with Logistic is expected since Logistic optimizes per-disease calibration independently.
>
> The structured VI posterior also shows tight concentration and robust pathways across PIP thresholds 0.6–0.9:
>
> | Evaluation Target | Metric | Result |
> |---|---|---|
> | Posterior Concentration | \# Active edges (95% CI) | 27.2 [25, 29] |
> | | Mean hyperedge size (95% CI) | 9.68 [9.28, 10.08] |
> | | Multi-pathway diseases (95% CI) | 86.4% [84.8%, 87.9%] |
> | Threshold Robustness | Disease-set Jaccard | 0.957 – 0.993 |
> | | Risk-factor-set Jaccard | 0.910 – 0.980 |
>
> The structural posterior is also predictively informative: retaining only high-PIP edges preserves essentially all AUC, while low-PIP edges perform near chance (see aTre-Q2 above for the full breakdown by prevalence group).
>
> **(2) Structural PIP calibration and MCMC comparison (simulation)**
> To validate the accuracy of the structural posterior, we compute PIP ECE for $H$, $z$, and $\gamma$ on the paper's simulation setting and compare against a full Gibbs sampler with Pólya-Gamma augmentation ($\mu = 1.5, \sigma = 0.5$, 50 replications):
>
> | Method | $H$ overlap | ECE$_H$ |   ECE$_z$ | ECE$_\gamma$  |  Predictive AUC |
> | ------ |  ----------: | --------: | --------: | --------: | --------: |
> | MCMC   |   0.885  ± 0.068 | 0.081  ± 0.019 |   0.058  ± 0.053 |  0.039  ± 0.032 | 0.747 ± 0.009 |
> | CAVI   |    0.850  ± 0.054 | 0.087  ± 0.026 | 0.028 ± 0.081 |   0.050 ± 0.030 | 0.741  ± 0.008 |
>
> These results show that CAVI and MCMC achieve comparable PIP calibration on the structural variables (ECE 0.03–0.09 for both methods), with MCMC performing slightly better. Structural recovery and predictive AUC are also closely matched. Taken together, these diagnostics support a more precise statement than the original text: **BHPI provides well-supported structural posterior uncertainty and competitive predictive calibration**.
>
>
> ### **Stronger structured baselines:**
>
> > **[W3/Q2]** baselines do not test whether hypergraph structure is better than other structured probabilistic latent-variable approaches...add at least one stronger structured baseline?
>
> We added **Bayesian Factor Regression (BFR)**: $y_{i,v} \sim \text{Bernoulli}(\sigma(\alpha_v + x_i^\top \Lambda f_v))$ with shared loading matrix $\Lambda$ (P×K), per-disease factors $f_v$ (K×1), and Normal priors. We report mean ΔAUC (×100) relative to per-disease Logistic with paired Wilcoxon p-values across 66 diseases. The clique-expanded baseline (see aTre-Q1) is included for comparison.
>
> | Prev. | BHPI | BFR K=20 | BFR K=30 | Clique-expanded | N |
> |---|---|---|---|---|---|
> | <2% | **+1.2** (3e-5) | +0.1 (0.29) | **−0.5** (1e-3) | **+0.1** (0.01) | 19 |
> | 2-5% | +0.2 (0.11) | **−0.3** (2e-6) | **−0.8** (2e-8) | **−0.2** (5e-4) | 28 |
> | ≥5% | **+0.2** (0.02) | **−0.2** (4e-6) | **−0.5** (4e-6) | **−0.3** (4e-6) | 19 |
>
> BHPI is the only method that consistently outperforms Logistic, with the advantage strongest for rare diseases (+1.2, p=3e-5). BFR and the clique-expanded baseline cannot match Logistic at any prevalence group — confirming that the benefit comes from hypergraph structure specifically, not Bayesian low-rank sharing or pairwise coupling.
>
>
> ### **Linear observation model:**
>
> > **[W2]** observation model remains linear...position BHPI relative to...Bayesian deep learning approaches that combine nonlinear predictors with uncertainty quantification
>
> **BHPI is designed to learn interpretable, overlapping, risk-modulated higher-order structure**. The linear observation model is a deliberate design choice that preserves pathway-level interpretability and enables closed-form CAVI updates. We thank the reviewer for pointing to Bayesian deep learning approaches as a reference point — combining nonlinear predictors with the hypergraph structure is an appealing extension that we will discuss in the camera-ready.

---

> > ### Author Rebuttal · Reviewer_eZ3K · 2026-04-03
> >
> > Thank you, this resolves my concerns, happy to raise my score to 5

---

> > > ### Author Response · Authors · 2026-04-04
> > >
> > > Thank you very much for being willing to raise the score. I just wanted to gently note that the score may not have been updated yet, in case that was unintentional. Thank you again for your time and consideration.

---

### Official Review · Reviewer_Lc2o · 2026-03-13

**Soundness:** 3
**Presentation:** 3
**Significance:** 3
**Originality:** 3
**Overall Recommendation:** 5
**Confidence:** 5

**Summary:**

This work models multi‑disease risk factors using electronic health records and developed Bayesian hypergraph inference framework. The method is developed and applied as well to real world data.

**Compliance With Llm Reviewing Policy:**

Affirmed.

**Final Justification:**

The authors clarified many unclear algorithmic details during the rebuttal, and many points are now much clearer for readers. It did change my view of the work, and I am happy to adjust my score. I acknowledge that I didn't fully check all the computational parts, e.g., eqns in Expectation (Section A4.1). The authors are strongly recommended to revise the algorithm description.

1. To list all latent variables in q(Z) decomposition with the conditional VI constraint. The missed terms should be added to avoid misleading.
2. To add a full PGM plate in the Appendix. (TikZ suggested. Circles for latent RV, shaded circle for Obs, squares for param, etc, are kindly suggested, following the PGM plot habit). The introduced latent variables in BHPI are kinda a lot, and this PGM part is quite necessary for easy following.
3. Introduce the non-conjugate situation and Polya-Gamma parameter $\omega$, ahead of the variational family equation in section 4.1. Currently, $\omega$ is introduced too late, which ruins readability.
4. The description in the Appendix that mentions Mean-field is contradicting, i.e., "we use the mean-field variational
inference (MFVI) (Blei et al., 2017), which assumes mutual independence across each parameter". Please replace it with the real claim, i.e., "structured VI scheme".

**Key Questions For Authors:**

1. How the mean-field VI is used in this work?
2. What are the obs, latent variables and PGM, and what is the joint prob, if variational inference method is designed?
3. How to define factor risk in your applications?

**Limitations:**

As detailed in last section, the methodology is poorly addressed.

**Strengths And Weaknesses:**

Soundness:

The joint probability distribution over the latent and observed variables is not explicitly defined, which undermines the derivation of the evidence lower bound (ELBO).
The resulting ELBO appears as a mere collection of symbols without a clear probabilistic foundation.
The absence of a probabilistic graphical model (PGM) plate makes it impossible for readers to understand how dependencies between observations and latent variables are structured. The model’s generative process is fairly ambiguous.
Third, the authors claim to develop "Structured variational inference" and avoid mean-field variational inference, yet in Appendix A4.1 they cite “we use the mean-field variational
inference (MFVI) (Blei et al., 2017) )” and appear to rely on its machinery.
This contradiction leads the readers to nowhere.
Fourth, the algorithm description in A4.1 stops at expectations and does not provide the iterative update formulas for variational parameters, this is weird.
The reader may notice that Algorithm in page 5 lists equations (9)-(15) for these updating rules. However, all these equations are about posterior pdf of latent variable, instead of iteration formulas.

Presentation:

The clarity of the paper requires substantial improvement. The mathematical derivations are overly dense and often confusing, making it difficult for readers to follow.
A clear inconsistency exists between Figure 1(which lists variational functions) and the description in Section A4.1.
Moreover, the experimental setup for the UK Biobank application is poorly described. How the authors derive the“risk factor pathway” information from the raw data, and how the data are linked to the model. All are uncertain.

Significance:

While modeling multi‑disease risk factors using electronic health records is an important problem, the current presentation obscures the potential impact of the proposed framework.
The lack of a clear probabilistic model and incomplete algorithmic details prevent the reader from assessing whether the method can deliver meaningful improvements in practice.

Originality:

The combination of structured variational inference with hypergraphs for modeling disease pathways represents a novel direction. However, the originality is difficult for a reader to sum-up  if from paper-reading.

---

> ### Author Rebuttal · Authors · 2026-03-31
>
> We thank the reviewer for the detailed critique. We recognize that the paper's appendix presentation made the probabilistic model harder to follow than intended. We organize our response by topic, writing out the key structure directly.
>
> ### **On the probabilistic model:**
>
> > **[Q2]** What are the obs, latent variables and PGM, and what is the joint prob, if variational inference method is designed?
>
> > **[W1-i]** The joint probability distribution is not explicitly defined...no PGM plate...generative process is fairly ambiguous
>
> Thank you for this question. The paper specifies the full probabilistic model:
> - the observation model in Eq. (1); the observations are risk factors $X$, and outcomes $Y$,
> - the latent variables $\Theta = \{z, r, m, \rho, \mu, \gamma, \nu, \sigma^2_\mu, \alpha\}$ in Sections 3.5, 3.6 and Appendix A.2,
> - and the posterior (Eqs. 4–5):
> $$p(\Theta \mid Y, \omega) \propto p(Y \mid \Theta, \omega) \prod_e p(z_e \mid r_e)p(r_e) \prod_{v,e} p(m_{v,e} \mid z_e, \rho_{v,e})p(\rho_{v,e}) \prod_j p(\gamma_j \mid z, H, \nu_j) \prod_{j,e} p(\mu_{j,e} \mid \gamma_{j,e}, \sigma^2_\mu)p(\nu_{j,e}) \cdot p(\sigma^2_\mu)$$
> where the $\gamma$ prior includes the repulsion penalty on overlapping hyperedges (Section 3.6, Eq. 5), $\omega$ are Pólya-Gamma augmentation variables, and $\beta_{j,v} = d_v^{-1}\sum_e H_{v,e}\mu_{j,e}$ (Eq. 2).
>
> Together, these define a complete generative model — the ELBO is derived from this specification, not the other way around.
> The generative DAG is: $z_e \to m_{v,e}$; $H_{v,e} = z_e m_{v,e}$; $\{z_e, H\} \to \gamma_{j,e} \to \mu_{j,e}$; $\beta_{j,v} = d_v^{-1}\sum_e H_{v,e}\mu_{j,e}$; $y_{i,v} \sim \text{Bernoulli}(\sigma(\alpha_v + x_i^\top \beta_v))$ (Sections 3.3–3.6, Eqs. 1–5). We will add a formal plate diagram in the camera-ready.
>
> ### **On the inference algorithm:**
>
> > **[Q1]** How the mean-field VI is used in this work?
>
> > **[W1-ii]** claim 'Structured variational inference' yet cite MFVI...This contradiction leads the readers to nowhere
>
> **BHPI's variational family is structured, not mean-field; the MFVI reference in A.3 is the general CAVI derivation framework, not the approximation family we use.** There we first wrote the **generic MFVI/CAVI derivation** for the standard ELBO update rule in Eq. (7), and then specialized it to the BHPI variational family through a conditional-entropy term in Eq. (8). The variational family explicitly contains conditional factors $ q(m_{v,e}\mid z_e), q(\gamma_{j,e}\mid z_e), q(\mu_{j,e}\mid \gamma_{j,e}) $, which preserve the logical dependencies emphasized in Section 4.1.
>
> > **[W1-iii]** does not provide the iterative update formulas...all equations are about posterior pdf...instead of iteration formulas
>
> Eqs. (9)-(15) in A.4 give the optimal coordinate-wise variational factors under CAVI. In implementation, the corresponding variational parameters are updated by substituting the current expectations of the other factors into these expressions at each iteration, as summarized in Algorithm 1. For example, the update for $q(z_e)$ takes current estimates of $q(m_{v,e})$ and $q(\gamma_{j,e})$ as inputs and outputs a closed-form Bernoulli parameter — exactly the substitution that Algorithm 1 executes at each coordinate step.
>
> > **[W2-i]** inconsistency exists between Figure 1...and...Section A4.1
>
> Figure 1 (bottom-left) summarizes the key structured variational family, whereas A4.1 provides the CAVI machinery used to optimize that family. These serve complementary roles; the dependency DAG above (Q2/W1-i) clarifies how they connect.
>
>
> ### **On the UKB setup and risk factors:**
>
> > **[Q3]** How to define factor risk in your applications?
>
> > **[W2-ii]** The experimental setup for the UK Biobank application is poorly described
>
> In UKB, the risk factors are the 71 observed baseline covariates (demographics, biomarkers, lifestyle, socioeconomic variables) described in Section 5.5 and Table 5 (Appendix C), and the outcomes are 66 chronic diseases. Their effects on disease risk are mediated through the latent hyperedges: $\gamma_{j,e}$ indicates whether risk factor $j$ acts on pathway $e$, $\mu_{j,e}$ gives the pathway-level effect size, and the induced disease-level effect is $\beta_{j,v} = d_v^{-1}\sum_e H_{v,e}\mu_{j,e}$. For a given patient $i$, the risk contribution of pathway $e$ to disease $v$ is $d_v^{-1} H_{v,e} \sum_j \mu_{j,e} x_{i,j}$, and the total risk is $\eta_{i,v} = \alpha_v + \sum_e (\text{pathway } e\text{'s contribution})$, as defined in Eqs. (1)–(2). The pathway structure and risk-factor assignments are learned by the model, not derived from the raw data in preprocessing.

---

> > ### Author Rebuttal · Reviewer_Lc2o · 2026-04-03
> >
> > The authors clarified many unclear algorithmic details during the rebuttal, and many points are now much clearer for readers. It did change my view of the work, and I am happy to adjust my score. I acknowledge that I didn't fully check all the computational parts, e.g., eqns in Expectation (Section A4.1). The authors are strongly recommended to revise the algorithm description.
> >
> > 1. To list all latent variables in q(Z) decomposition with the conditional VI constraint in \textbf{Fig. 1}. The missed terms should be added to avoid misleading.
> > 2. To add a full PGM plate in the Appendix. (TikZ suggested. Circles for latent RV, shaded circle for Obs, squares for param, etc, are kindly suggested, following the PGM plot habit). The introduced latent variables in BHPI are kinda a lot, and this PGM part is quite necessary for easy following.
> > 3. Introduce the non-conjugate situation and Polya-Gamma parameter $\omega$, ahead of the variational family equation in section 4.1. Currently, $\omega$ is introduced too late, which ruins readability.
> > 4. The description in the Appendix that mentions Mean-field is contradicting, i.e., "we use the mean-field variational
> > inference (MFVI) (Blei et al., 2017), which assumes mutual independence across each parameter". Please replace it with the real claim, i.e., "structured VI scheme".

---

> > > ### Author Response · Authors · 2026-04-04
> > >
> > > We sincerly thank you for the thoughtful re-evaluation and constructive suggestions. We will further clarify the algorithm description in the camera-ready, incorporating all four suggestions.

---

### Official Review · Reviewer_aTre · 2026-03-13

**Soundness:** 3
**Presentation:** 3
**Significance:** 4
**Originality:** 3
**Overall Recommendation:** 5
**Confidence:** 4

**Summary:**

The authors argue that while multi-task learning methods applied to Electronic Health Records (EHR) aim to learn shared representations across multiple diseases, they are inherently black-box models that limit the ability to identify specific groups of related diseases and do not provide uncertainty estimates. To address this limitation, the authors propose Bayesian Hypergraph Pathway Inference (BHPI), a framework that discovers latent disease pathways through hypergraph structure learning. The proposed model incorporates a repulsion prior to prevent redundant hyperedges (i.e., highly overlapping pathways) and performs posterior inference via a structured variational inference approach. Through experiments on simulated data and the UK Biobank dataset, the authors demonstrate the predictive performance of the proposed method as well as its ability to disentangle risk-factor–specific disease pathways.

**Compliance With Llm Reviewing Policy:**

Affirmed.

**Final Justification:**

The authors effectively addressed my remaining concerns during the rebuttal period, which reinforces my previous positive assessment.

**Key Questions For Authors:**

1. To better demonstrate the validity of utilizing a hypergraph structure, it would be helpful to compare the proposed approach with a clique-expanded graph representation combined with a GCN. In particular, the authors could construct a graph using the same discovered hyperedges and then apply clique expansion to convert it into a pairwise graph. If the proposed model shows significantly better performance than such a baseline, it would more clearly demonstrate the advantage of modeling higher-order interactions beyond pairwise relationships.

2. The qualitative analysis presented in the Pathway-level Illustration section shows promising results. However, the evaluation remains largely qualitative. It would strengthen the paper if the authors provide quantitative analysis of the discovered pathways. For example, the authors could evaluate predictive performance using (1) only highly confident edges (high PIP) and (2) only uncertain edges (low PIP). Such an experiment would help verify whether the uncertainty estimates correspond to meaningful structural information.

3. The authors should provide code and other necessary materials to facilitate reproduction of the reported results.

**Limitations:**

yes

**Strengths And Weaknesses:**

* Soundness: The proposed approach is methodologically sound. Utilizing Bayesian modeling to place priors on pathway existence and disease membership is reasonable and well-motivated. In addition, modeling disease pathways with a hypergraph structure is appropriate, since pathways naturally involve higher-order relationships that cannot be adequately represented by pairwise interactions.

* Presentation: Despite the complexity of the Bayesian modeling framework, the presentation of the paper is relatively clear and easy to follow.

* Significance: Beyond predicting diseases, discovering disease pathways is highly important for advancing biological and medical research. However, additional evaluation would be beneficial to further verify the biological validity of the discovered pathways.

* Originality: The paper proposes a novel modeling framework for discovering latent disease pathways from EHR data, which demonstrates a clear level of originality.

---

> ### Author Rebuttal · Authors · 2026-03-31
>
> We thank the reviewer for the positive assessment of the formulation, clarity, and significance, and for the concrete suggestions on how to strengthen the empirical case.
>
> > **[Q1]** compare the proposed approach with a clique-expanded graph representation combined with a GCN...demonstrate the advantage of modeling higher-order interactions beyond pairwise relationships
>
> Thank you for this excellent suggestion. We implemented the requested comparison using the **same discovered hyperedges**, converted to a clique-expanded disease graph. The per-disease ΔAUC (×100) relative to per-disease Logistic comparison (including clique and Bayesian Factor Regression) is in our response to eZ3K-W3/Q2 below. On UKB, BHPI achieves a significant ΔAUC of +1.2 over Logistic for rare diseases (p=3e-5); the best clique-expanded pairwise representation variant achieves only +0.1. A learnable clique-GCN also failed (AUC ≈ 0.50). The main reason is representational: on UKB the clique-expanded graph is ~65% dense even at PIP threshold 0.9, so clique expansion collapses higher-order group membership into near-uniform pairwise links. The primary advantage of the hypergraph model is therefore **structural selectivity**: BHPI identifies which diseases co-participate in the same risk-modulated pathway — an assignment that clique expansion cannot recover by construction.
>
> > **[Q2]** evaluate predictive performance using (1) only highly confident edges (high PIP) and (2) only uncertain edges (low PIP)...verify whether the uncertainty estimates correspond to meaningful structural information.
>
> Thank you for this insightful suggestion. We added this analysis in both UKB and simulation.
>
> | Configuration | <2% (n=19) | 2-5% (n=28) | ≥5% (n=19) | Simu |
> |--------|------------|-------------|------------|--------------|
> | Full model | 0.689 | 0.697 | 0.716 | 0.757  |
> | **High-PIP only** | 0.687 | 0.696 | 0.716  | 0.756 |
> | Random subset | 0.602 | 0.613 | 0.627 | 0.657 |
> | Low-PIP only | 0.586 | 0.550 | 0.574 | 0.512 |
>
> Thus, the high-PIP edges retain essentially all predictive signal, while low-PIP edges are close to chance. This confirms that the posterior structure carries meaningful information rather than reflecting arbitrary uncertainty.
>
> > **[Q3]** The authors should provide code and other necessary materials to facilitate reproduction of the reported results.
>
> A complete implementation along with the simulation reproduction scripts is available at the anonymous repository included in the supplementary materials submitted with the paper.
>
> > **[W1]** additional evaluation would be beneficial to further verify the biological validity of the discovered pathways
>
> We appreciate this suggestion. The manuscript's pathway examples align with established clinical literature. The smoking-related respiratory pathway is consistent with the well-documented smoking–respiratory disease association (Forey et al., 2011); the separate mucosal damage pathway aligns with smoking as a peptic-ulcer risk factor (Kurata & Nogawa, 1997); and the three dementia-related pathways are supported by the vascular cognitive impairment literature (van der Flier et al., 2018), frailty–dementia clustering (Kojima et al., 2016), and the diabetes–dementia link (Xue et al., 2019). Independent multimorbidity clustering studies also report closely related respiratory, cardiometabolic, neuropsychiatric, and dementia-linked patterns (Grande et al., 2021; Simões et al., 2017). We view these as strong plausibility checks. The PIP-stratified analysis (Q2 above) provides direct quantitative support: high-confidence edges carry essentially all predictive signal, while low-confidence edges do not.
>
> **References**
>
> [1] Forey et al. (2011). Systematic review with meta-analysis of the epidemiological evidence relating smoking to COPD, chronic bronchitis and emphysema. *BMC Pulm Med*.
>
> [2] Kurata & Nogawa (1997). Meta-analysis of risk factors for peptic ulcer. *J Clin Gastroenterol*.
> [3] van der Flier et al. (2018). Vascular cognitive impairment. *Nat Rev Dis Primers*.
>
> [4] Kojima et al. (2016). Frailty as a Predictor of Alzheimer Disease, Vascular Dementia, and All Dementia Among Community-Dwelling Older People. *JAMDA*.
>
> [5] Xue et al. (2019). Diabetes mellitus and risks of cognitive impairment and dementia. *Ageing Res Rev*.
>
> [6] Grande et al. (2021). Multimorbidity burden and dementia risk in older adults. *Alzheimer's Dement*.
>
> [7] Simões et al. (2017). Patterns and Consequences of Multimorbidity in the General Population. *Arthritis Care Res*.

---

> > ### Author Rebuttal · Reviewer_aTre · 2026-04-06
> >
> > I believe that the methods proposed in this paper are sound, and the experiments on predictive performance for each edge with different confidence levels add significant strength to the paper.

---

> > > ### Author Response · Authors · 2026-04-07
> > >
> > > Thank you for your constructive feedback and for the positive assessment of our work. We will incorporate these new experimental results into the final version of the manuscript.

---

### Decision · Program_Chairs · 2026-04-30

**Decision:**

Accept (spotlight)

**Comment:**

This paper introduces the notion of latent hypergraphs to represent overlapping disease pathways, which is a novel and principled departure from pairwise graphs and independent-disease models that are currently in use. All reviewers were impressed by originality and sophistication of the proposed framework and found its components to be well justified by the application. As one reviewer said, the "repulsion prior is well motivated" and addresses "exactly the kind of failure mode one would worry about in these models." The three main initial concerns of reviewers of uncertainty calibration, baseline adequacy, and presentation clarity were all adequately addressed during the discussion period. The paper still has one main limitation of a linear observation model, but this is a defensible design choice for interpretability. All four reviewers were eager to accept. I think this paper serves a great exemplar of application-inspired ML: a highly original idea, well-tailored to an important and timely application, with careful empirical work substantiating the approach's effectiveness.